# NR2F2 regulation of interstitial cell fate in the embryonic mouse testis and its impact on differences of sex development

Martín Andrés Estermann [1], Sara A. Grimm [2], Abigail S. Kitakule[1], Karina F. Rodriguez[1], Paula R. Brown[1], Kathryn McClelland[1], Ciro M. Amato[1,3] & Humphrey Hung-Chang Yao [1] ✉

Testicular fetal Leydig cells produce androgens essential for male reproductive development. Impaired fetal Leydig cell differentiation leads to differences of sex development including hypospadias, cryptorchidism, and infertility. Despite fetal Leydig cells are thought to originate from proliferating progenitor cells in the testis interstitium, the precise mechanisms governing the interstitial cells to fetal Leydig cell transition remain elusive. Using mouse models and single-nucleus multiomics, we find that fetal Leydig cells arise from a *Nr2f2*-positive interstitial population. Embryonic deletion of *Nr2f2* in mouse testes results in differences of sex development, including dysgenic testes, Leydig cell hypoplasia, cryptorchidism, and hypospadias. By combining single-nucleus multiomics and NR2F2 ChIP-seq we find that NR2F2 promotes the progenitor fate while suppresses Leydig cell differentiation by modulating key transcription factors and downstream genes. Our findings establish *Nr2f2* as a crucial regulator of fetal Leydig cell differentiation and provide molecular insights into differences of sex development linked to *Nr2f2* mutations.

Testes are a pair of oval-shaped male reproductive organs with dual roles; they produce sperm, the male gametes, and androgens, the primary masculinizing hormones. Sperm production occurs within the seminiferous tubules, where germ cells are supported by Sertoli cells, providing nourishment and a protective environment. Surrounding the seminiferous tubules is the interstitial space or interstitium, which contain various cell types including steroidogenic fetal Leydig cells and non-steroidogenic interstitial cells[1–3]. While much research has focused on Sertoli and germ cells due to their direct involvement in spermatogenesis and reproduction, the other testicular cells have often been overlooked.

Fetal Leydig cells, the steroidogenic cells within the interstitium, play a crucial role in synthesizing and secreting androgens, the hormones critical for the establishment of male reproductive tracts and appearance of male secondary sex characteristics[4,5]. Despite their

identification over a century ago[6,7], the origin and differentiation of fetal Leydig cells remain subjects of ongoing debate. It is widely accepted that fetal Leydig cells differentiate from proliferating progenitor cells within the testis interstitium, being detected in the embryonic mouse testis from embryonic day or E12.5 onwards[8–10]. However, the specific interstitial population giving rise to the fetal Leydig cell and the mechanism underlying the differentiation from interstitial to fetal Leydig cells is still not fully understood.

The hedgehog signaling pathway was the first pathway to be implicated in fetal Leydig cell differentiation. Desert hedgehog (DHH), secreted from Sertoli cells, binds to its receptor PTCH1 on interstitial cells, activating the GLI family of transcription to induce fetal Leydig cell differentiation[11–16]. Conversely, Notch[17,18] and VEGF[19] signaling pathways inhibit Leydig cell differentiation, maintaining the interstitial cell identity. Failure in fetal Leydig cell differentiation results in various

[1]Reproductive Developmental Biology Group, National Institute of Environmental Health Sciences, Research Triangle Park, Durham, NC, USA. [2]Integrative Bioinformatics, National Institute of Environmental Health Sciences, National Institutes of Health, Research Triangle Park, Durham, NC, USA. [3]Present address: Department of Surgery, Division of Urology, University of Missouri, Columbia, MO, USA. ✉e-mail: humphrey.yao@nih.gov

differences of sex development (DSDs) or conditions affecting male reproduction, including ambiguous genitalia, hypospadias, cryptorchidism and infertility[20–23].

Beyond fetal Leydig cells, the testicular interstitium contains immune cells, fibroblasts, vascular cells, and myoid cells among others, all of which play supportive roles in maintaining the homeostasis and functionality of the testicular microenvironment[1,3,10,24]. Little is known about their specific roles in testicular function and how they contribute to fertility. Non-steroidogenic interstitial cells are characterized by the expression of markers such as *Tcf21, Wnt5a, Arx, Pdgfra*, and *Nr2f2*[10,25–28]. While the non-steroidogenic interstitial cell population is generally considered homogeneous, it remains unclear whether these markers are uniformly expressed across different interstitial cell subtypes. *Tcf21*-positive (also known as *Pod1*)[25], as well as *Wnt5a*-positive[10] cells contributed to different interstitial somatic lineages, including fetal Leydig and peritubular myoid cells. Global *Tcf21* deletion in mice resulted in an expansion of fetal Leydig cell population[29] whereas *Arx*[26] and *Pdgfra*[28] knockout testes had reduced fetal Leydig cells number. In addition, *Tmsb10* expression in fetal Leydig cell progenitors was required for proper fetal Leydig cell differentiation[30]. These observations imply that fetal Leydig cell differentiation hinges on the balance between inhibitory and stimulatory factors.

*Nr2f2* (or COUP-TFII), an orphan nuclear receptor expressed in the gonadal interstitium, was implicated in adult Leydig cell differentiation in mice[31] and hypogonadism and DSDs in humans[32]. In this study, we comprehensively characterized the cellular composition of embryonic mouse testis, focusing on the diversity of interstitial cell populations. Furthermore, using multi-omics approaches and genetic mouse models we identified fetal Leydig cell progenitors and elucidated the molecular process governing the non-steroidogenic interstitial cells to fetal Leydig cell differentiation, with a specific emphasis on the role of *Nr2f2* in this transition.

## Results
### Characterizing the interstitial cell populations of embryonic testis using single-nucleus multiomics

To characterize the testicular cellular composition, and in particular, to gain insight into the poorly characterized interstitial cell diversity in mice, we performed combined single-nucleus RNA-seq and ATAC-seq (10X multiome) in E14.5 testes, a stage when somatic populations are already established. We obtained 7122 nuclei that were classified into 14 different cell clusters (c0 to c13) based on the combined transcriptomic and chromatin profiles (Supplementary Data 1). Cluster 13 only contained two cells and was excluded from further analysis. We used Manifold Approximation and Projection (UMAP) dimensional reduction to visualize the cell clusters in 2 dimensions based on their similarities and differences in the combined RNA-seq and ATAC-seq profiles (Fig. 1A). Cell clusters were assigned based on the expression of known gonadal cell markers (Supplementary Fig. 1 and *Methods*), including Sertoli cells (c2), fetal Leydig cells (c3), germ cells (c4 and c6), supporting-like cells (c9), endothelial (c11 and c12) and immune cells (c8) (Fig. 1B & Supplementary Fig. 1)[1].

The remaining cell clusters (c0, c1, c5, c7 and c10) gathered into a larger group in the UMAP, indicating a more similar transcriptome and/or chromatin profile (Fig. 1A). As these clusters expressed the known interstitial cell markers, including *Arx, Pdgfra, Wnt5a, Tcf21* and *Nr2f2* (Supplementary Fig. 2A), they were classified as potential interstitial cells. Despite their similarity, these five clusters were distinguished by unique combinations of marker expression: c0 with *Adam23* and *Inhba*, c1 with *Mylk, Myh11* and *Acta2*, c5 with *Aldh1a2, Pdgfc* and *Podxl*, c7 with *Itga8, Ntrk3* and *Rxfp2*, and c10 with *Lgi1, Dgkb* and *Ndst3* (Fig. 1C & Supplementary Fig. 1). To localize these cell populations in the testes, we performed immunofluorescence on E14.5 testis (Fig. 1D). Cluster 0 had the highest expression of *Inhba*, which is

expressed in steroidogenic cell progenitors[33]. Cluster 1 marker ACTA2 was expressed in the layer of cells directly underneath the testicular epithelium (tunica albuginea) and around the seminiferous tubules (peritubular myoid cells), suggesting a contractile cell cluster (Fig. 1D). Based on ALDH1A2 expression, c5 was classified as testicular epithelium (Fig. 1D). This cluster also expresses *Podxl*, recently identified to be expressed in fetal human gonadal epithelial cells, consistent with our findings[34]. Cells positive for ITGA8 (c7) were found in the mesonephros and in the gonadal region adjacent to the mesonephros, suggesting a mesonephric mesenchyme cluster (Fig. 1D). Lastly, expression of LGI1, which marks c10, was sparsely detected in the interstitium (Fig. 1D). This is consistent with the low abundance of cells in the single-nucleus sequencing results (less than 1%). This cluster (c10) was classified as pericytes, as they also expressed high levels of testicular pericytes genes, including *Mcam, Steap4* and *Cspg4* (NG2)[19,35].

### Identification of potential fetal Leydig cell progenitors

To understand the relationship among the interstitial cell clusters, we evaluated the differences in chromatin accessibility and transcriptomic profiles of the different cell clusters. We generated a second UMAP, where nuclei were projected based only on differences or similarities in chromatin accessibility from the ATAC-seq data (Fig. 1E). In contrast to the combined RNA-seq/ATAC-seq UMAPs (Fig. 1A), fetal Leydig cells (c3) clustered closely to the c0 (potential steroidogenic progenitors) in the ATAC-seq UMAP (Fig. 1E). To assess the chromatin similarity between c3 and c0, we calculated the Euclidean distance between each cluster and generated a dendrogram of clusters from the distance matrix (Supplementary Fig. 2B). Since we cannot infer directly from the dendrogram which cluster (c0, c1, c5 or c7) is closest to c3, we reported the Euclidean distance calculated between c3 and each of the other clusters. Such analysis revealed that c0 is the cluster most similar to c3 (Supplementary Fig. 2B). These observations not only indicated that the chromatin changes during fetal Leydig cell differentiation were less abrupt than the transcriptomic changes, but also implicated c0 as potential fetal Leydig cell progenitors. To assess if c0 cluster represents the progenitor cells, we performed trajectory analysis based on the single nuclei ATAC-seq data (Supplementary Fig. 2C). Trajectory analysis showed that c0 and c3 are connected, and c0 is developmentally related to c3 (Supplementary Fig. 2C). To characterize this process of progenitors (c0) to fetal Leydig cell (c3) differentiation, we compared the transcriptomic changes between c0 and c3 (Fig. 1F). As expected, fetal Leydig cells expressed several steroidogenic genes, including *Cyp17a1, Cyp11a1, Hsd3b1, Nr5a1* (steroidogenic factor 1) and *Insl3* (Fig. 1F). Pathway analysis of this cluster revealed higher expression of lipid and cholesterol metabolism genes (Supplementary Fig. 2D, Supplementary Data 2). Cluster 0, on the other hand, expressed genes involved in extracellular matrix organization and collagen formation, indicative of mesenchymal cell characteristics (Supplementary Fig. 2E, Supplementary Data 2) and several interstitial cell markers, including *Tcf21* (Pod1), *Arx, Pdgfra, Tmsb10* and *Nr2f2*. The orphan nuclear receptor *Nr2f2* drew our attention for its role in adult Leydig cell differentiation and potential link to human DSDs[31,32].

Lastly, to determine whether the differentiation of progenitor cells (c0) to fetal Leydig cell (c3) process also occurs at early developmental stages, we integrated the E14.5 multiomic dataset from this study with datasets from E12.5 and E13.5 embryonic testis (Supplementary Fig. 2F)[36,37]. Clustering and expression analysis of the combined dataset identified c2 as steroidogenic progenitors (*Nr2f2* positive) and c10 as fetal Leydig cells (*Nr5a1* positive) (Supplementary Fig. 2F, G). Moreover, *Nr2f2* positive steroidogenic progenitor cells form a continuous trajectory with c10 fetal Leydig cells across all evaluated timepoints (E12.5, E13.5 and E14.5) (Supplementary Fig. 2F). These findings suggest a conserved differentiation program from

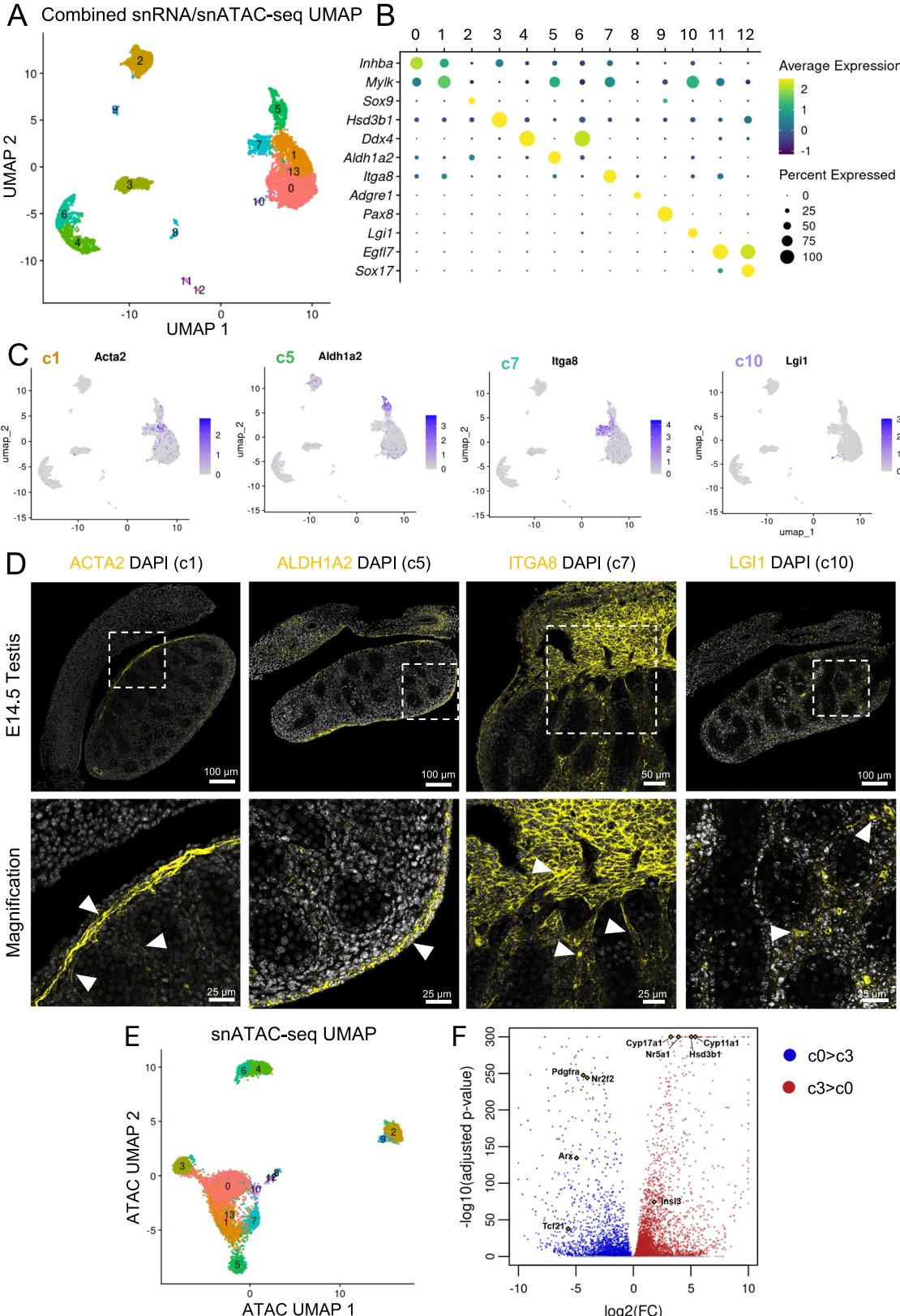

**Fig. 1 | Single-nucleus multiomics of E14.5 mouse testes. A** UMAP projection of the single-nucleus multiomics (combined RNA-seq and ATAC-seq) data from E14.5 testes, color-coded by cell clusters. **B** Dot plot representation of the expression of cell specific marker genes for each cluster. **C** Feature plots of the normalized expression of interstitial genes on the multiomic UMAP visualization of E14.5 testes. **D** Immunofluorescence for cluster/cell type specific markers (Yellow) in E14.5 male testes. Samples were counterstained with DAPI (Gray). White arrowheads indicate immunopositive cells. White dashed box indicates the magnified area. **E** UMAP of the single-nucleus ATAC-seq data from E14.5 testes, color-coded based on the multiomic (combined RNA-seq and ATAC-seq) cell clusters. **F** Volcano plot of differentially expressed genes between c0 (steroidogenic progenitor cells) and c3 (fetal Leydig cells). Blue and red dots represent significantly upregulated genes in c0 or c3 respectively.

*Nr2f2* positive steroidogenic progenitor cells to fetal Leydig cells throughout early testicular development.

## NR2F2 is expressed in non-steroidogenic interstitial cells but not in fetal Leydig cells

To investigate the role of *Nr2f2* during testicular differentiation, we performed immunofluorescence for NR2F2 and other somatic cell markers in the beginning of testis differentiation at E11.5, the appearance of fetal Leydig cells at E12.5-E14.5, and complete morphogenesis of the testis at E17.5 (Fig. 2A). At E11.5 (Fig. 2A), most of the somatic cells inside the undifferentiated testis and a few cells in the coelomic epithelium that delimits the fetal testis were positive for the transcription factor NR5A1 as previously reported[38]. On the other hand, NR2F2 was mainly localized to NR5A1-negative somatic cells in the coelomic epithelium and the mesonephros that attach to the testes. At E12.5 when morphogenesis of fetal testis occurs and testicular cords form, NR2F2 was detected now in the coelomic epithelium and the interstitial space between the NR5A1-positive Sertoli cells in testicular cords (Fig. 2A). This expression pattern was maintained through the different developmental stages (E13.5, E14.5 and E17.5) (Fig. 2A). Consistent with the single-nucleus RNA-seq data, NR2F2 was not present in the NR5A1-positive fetal Leydig cells in the interstitium at E14.5 and E17.5 (Fig. 2A). Interestingly, a few NR2F2 and NR5A1 double positive cells were detected in the gonadal interstitium, expressing lower levels of both markers (white arrowheads) (Fig. 2A). They could potentially be steroidogenic progenitor cells differentiating into fetal Leydig cells.

To further characterize the NR2F2+ interstitial cell types, we performed immunofluorescence on E14.5 testes for different testicular cell markers (Fig. 2B). In corroboration with the single-nucleus RNA-seq data, NR2F2 was not expressed in AMH+ Sertoli cells, CYP17A1+ steroidogenic Leydig cells, nor PECAM1+ vascular endothelial cells. In contrast, NR2F2 colocalized with two known interstitial cell markers: the transcription factor ARX[26] and the membrane receptor PDGFRA[28]. Tunica albuginea cells, positive for smooth muscle actin (ACTA2), were also positive for NR2F2. The single-nucleus multiomic data and the immunofluorescent results together indicate that the testis interstitium is composed of NR2F2- steroidogenic fetal Leydig cells and heterogenous NR2F2+ non-steroidogenic cells.

## *Nr2f2*-positive interstitial cells differentiate into fetal Leydig cells

The single-nucleus multiomic data suggested that NR2F2+ interstitial cells could be the progenitors of NR2F2- steroidogenic fetal Leydig cells (Fig. 1E, Supplementary Fig. 2B, C). To test this hypothesis, we developed a tamoxifen-inducible lineage tracing *Nr2f2*-CreER mouse model[39] (Fig. 3A). When the *Nr2f2*-CreER mice were crossed to the *CAG-Sun1/sfGFP* reporter mice, upon exposure to tamoxifen, the *Nr2f2*+ cells in the resulting embryos become permanently labeled with perinuclear GFP. Pregnant females carrying *Nr2f2*-CreER; *CAG-Sun1/sfGFP* embryos were injected with tamoxifen before (E11.5), during (E12.5) or after (E13.5 and E14.5) the appearance of fetal Leydig cells (Fig. 3). The fate of the *Nr2f2* + (*Sun1-GFP* +) cells in the fetal testes was evaluated at E14.5 or E17.5 (Fig. 3), showing widespread levels of GFP in the gonadal interstitium and mesonephros (Supplementary Fig. 3A). When *Nr2f2*+ cells were labeled at E11.5 (Fig. 3B), GFP was detected in NR2F2-positive cells (white arrowheads), implying that some *Nr2f2*+ cells at E11.5 continue to express NR2F2 at E14.5 (Fig. 3B). Perinuclear GFP was also detected in NR2F2-negative cells (yellow arrowheads), indicating that these cells expressed *Nr2f2* at E11.5 and then downregulated *Nr2f2* expression and differentiated into other cell types by E14.5 (Fig. 3B). To identify whether these GFP+/NR2F2- negative cells are fetal Leydig cells, we performed immunofluorescence for NR5A1 (or Steroidogenic factor 1/SF1), which labels Sertoli cells weakly in the testis cords and fetal Leydig cells strongly in the interstitium, or HSD3B1, which is only expressed in fetal Leydig cells (Fig. 3B). Some

NR5A1-positive cells in the interstitium, but not in testis cords, were positive for GFP (yellow arrowheads). In addition, a fraction of HSD3B1-positive fetal Leydig cells were also positive for GFP (yellow arrowheads), indicating that fetal Leydig cells derive from a *Nr2f2* positive interstitial cell lineage (Fig. 3B). Moreover, no GFP was detected in the AMH+ Sertoli cell lineage, indicating that interstitial cells and Sertoli cells are different cell lineages at E11.5 (Fig. 3B).

Lineage tracing of E12.5 (Fig. 3C) and E13.5 (Fig. 3D) *Nr2f2*-positive cells also labeled both non-steroidogenic interstitial (NR2F2+, white arrowheads) and fetal Leydig cells (interstitial NR5A1+, yellow arrowheads), but not Sertoli cells (weak NR5A1 staining in testis cords) (Fig. 3C, D). This finding implied that interstitial to Leydig lineage differentiation does not occur in a specific developmental time point, but rather is a continuous process during testicular development. We performed additional lineage tracing of *Nr2f2* positive cells at E14.5 and evaluated the fate of the cells at E17.5 (Fig. 3E). Consistent with our previous results, we detected GFP-positive interstitial (white arrowheads) and fetal Leydig cells (yellow arrowhead), but not Sertoli cells (Fig. 3E), supporting the model for a continuous Leydig cell differentiation from the *Nr2f2*-positive interstitial progenitor pool and confirming the steroidogenic progenitor to fetal Leydig cell differentiation seen in the single nucleus ATAC-seq data.

## *Nr2f2* conditional knockout resulted in dysgenic testes and Leydig hypoplasia

To evaluate the role of NR2F2 in testis differentiation, we generated an inducible, conditional knockout of *Nr2f2* in the progenitor cells (*WT1creER; Nr2f2^{f/f}*, or cKO thereafter) before the emergence of the Leydig cell population. To ensure that *WT1creER* mouse model targets all *Nr2f2* positive cells in fetal testes, we produced pregnant females that carry *WT1creER; CAG-Sun1/sfGFP* reporter embryos. By injecting tamoxifen at E11.5 and E12.5, we labeled the *Wt1*+ progenitor cells with *Sun1-GFP* and examined colocalization of Sun1-GFP and NR2F2 at E14.5 and E17.5 (Supplementary Fig. 3B). Almost all NR2F2 positive cells were positive for Sun1-GFP, indicating that *WT1creER* can be used to efficiently target *Nr2f2* expressing cells (Supplementary Fig. 3B). We therefore generated pregnant females that carried *Nr2f2* conditional knockout (cKO) (*WT1creER; Nr2f2^{f/f}*) and control (*Nr2f2^{f/f}*) embryos. Tamoxifen was injected at E11.5 and E12.5 and testes were collected at E14.5 for bulk RNA-seq and phenotypic analyses. Principal component analysis (PCA) of the testis transcriptomes showed a clear separation between control and *Nr2f2* knockout samples (Fig. 4A). Differential expression analysis identified 1919 differentially expressed genes (adj p < 0.05), which included 878 upregulated genes and 1041 downregulated genes between cKO and control testes (Supplementary Data 3). The downregulated genes included interstitial cell markers *Pdgfra*, *Arx*, *Tmsb10* and *Tcf21* and steroidogenic genes *Cyp17a1*, *Cyp11a1* and *Hsd3b1* (Fig. 4B). Pathway analysis on downregulated genes (Supplementary Fig. 3C and Supplementary Data 3) revealed that genes involved in muscle contraction and extracellular matrix were among the most downregulated. This is consistent with the expression of NR2F2 in both steroidogenic progenitor cells and contractile cells in the testis (Fig. 2B). Pathway analysis of the upregulated genes in the cKO testis did not show any significant result (adj p < 0.05; Supplementary Fig. 3D and Supplementary Data 3).

To characterize the phenotypes in detail, we performed immunofluorescence of testicular cell specific markers on E14.5 control and *Nr2f2* cKO testes (Fig. 4C). The *Nr2f2* floxed allele has a LacZ reporter cassette inserted behind the second LoxP site, which is only expressed under the control of the *Nr2f2* promoter, upon deletion by the Cre recombinase[40]. To validate the efficiency of the tamoxifen induced recombination, we performed immunofluorescence to detect NR2F2 and LacZ, together with PDGFRA, a marker expressed in *Nr2f2* positive interstitial cells. This allowed us to detect deleted (LacZ + /NR2F2-) or non-deleted (LacZ-/NR2F2 + ) cells. As expected, all PDGFRA-positive

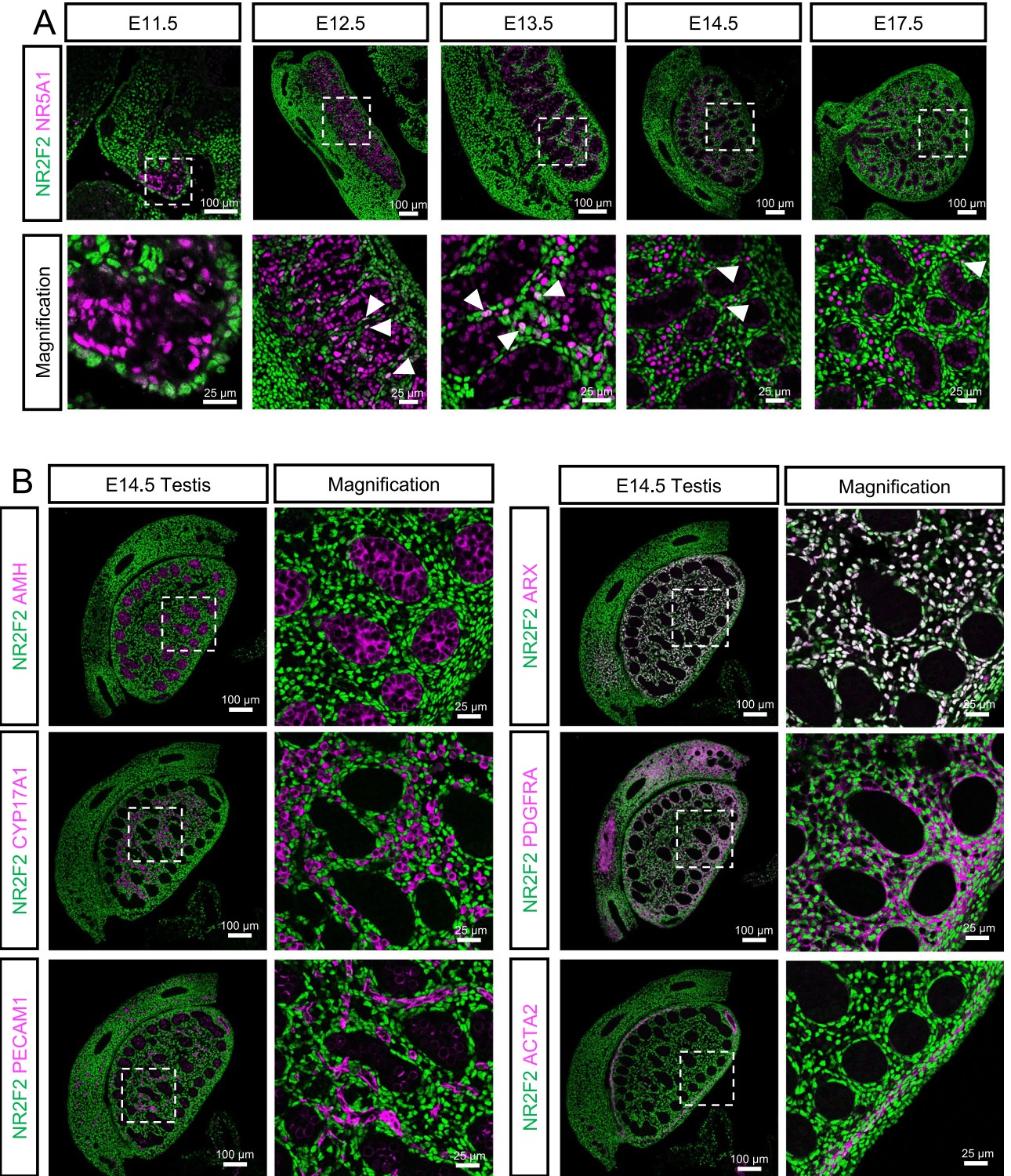

**Fig. 2 | NR2F2 protein localization during fetal testis differentiation.**
**A** Immunofluorescence for NR2F2 (green) and NR5A1 (magenta) in transverse
(E11.5) or longitudinal (E12.5, E13.5 E14.5 and E17.5) testis sections. White arrow-
heads indicate double positive cells for NR2F2 and NR5A1. White dashed box
indicates the magnified area. **B** Immunofluorescence of NR2F2 (green) and cell-type
specific markers (magenta, AMH for Sertoli cell, CYP17A1 for fetal Leydig cells,
PECAM1 for endothelial cells, ARX for interstitial cells, PDGFRA for interstitial cells,
and ACTA2 for contractile cells) in E14.5 male testes. White dashed box indicates
the magnified area.

interstitial cells in the control testes were positive for NR2F2. In the
cKO testes, instead, only a proportion of PDGFRA positive cells were
positive with NR2F2 (red arrowhead in Fig. 4C) whereas the rest of the
cells were LacZ positive and NR2F2 negative (blue arrowhead in
Fig. 4C). This suggests *Nr2f2* knockout occurred in a proportion of cells

as a result of incomplete action of the inducible Cre models. In few
instances, NR2F2 and LacZ were co-expressed in the same cell (white
arrowhead in Fig. 4C), which suggests a single allele recombination.
ARX, another marker of the non-steroidogenic interstitial cells, was
also expressed ubiquitously in the *Nr2f2* knockout testicular

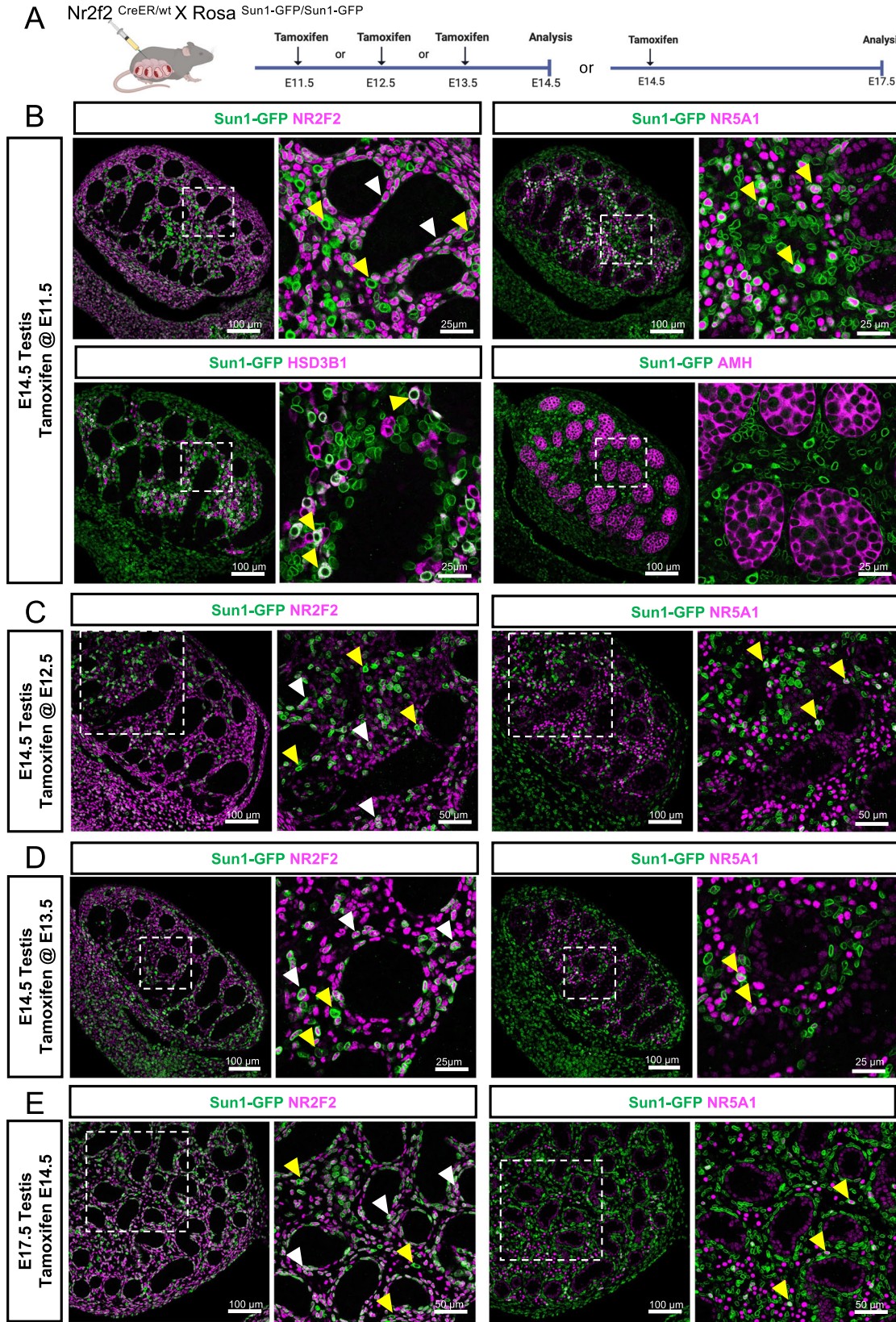

**Fig. 3 | Lineage tracing of *Nr2f2*-positive cells in the mouse fetal testis.**
**A** Schematic representation of the lineage tracing experiment of the fetal testis-derived *Nr2f2*+ cells in the *Nr2f2*-CreER; *CAG-Sun1/sfGFP* embryos. Created in BioRender, Yao, H. (2025) https://BioRender.com/ygb4q06. Cre activity was induced by tamoxifen administration at E11.5 (**B**), E12.5 (**C**), E13.5 (**D**) or E14.5 (**E**). Testes were collected at E14.5 (**B**–**D**) or E17.5 (**E**) and analyzed with immunofluorescence for peri-nuclear Sun1-GFP (green) and NR2F2, NR5A1, HSD3B1 or AMH (magenta). White arrowheads indicate NR2F2-positive cells labeled with Sun1-GFP. Yellow arrowheads indicate NR2F2-negative, Sun1-GFP-positive or Sun1-GFP/NR5A1 or Sun1-GFP/HSD3B1 double positive fetal Leydig cells. White dashed box indicates the magnified area.

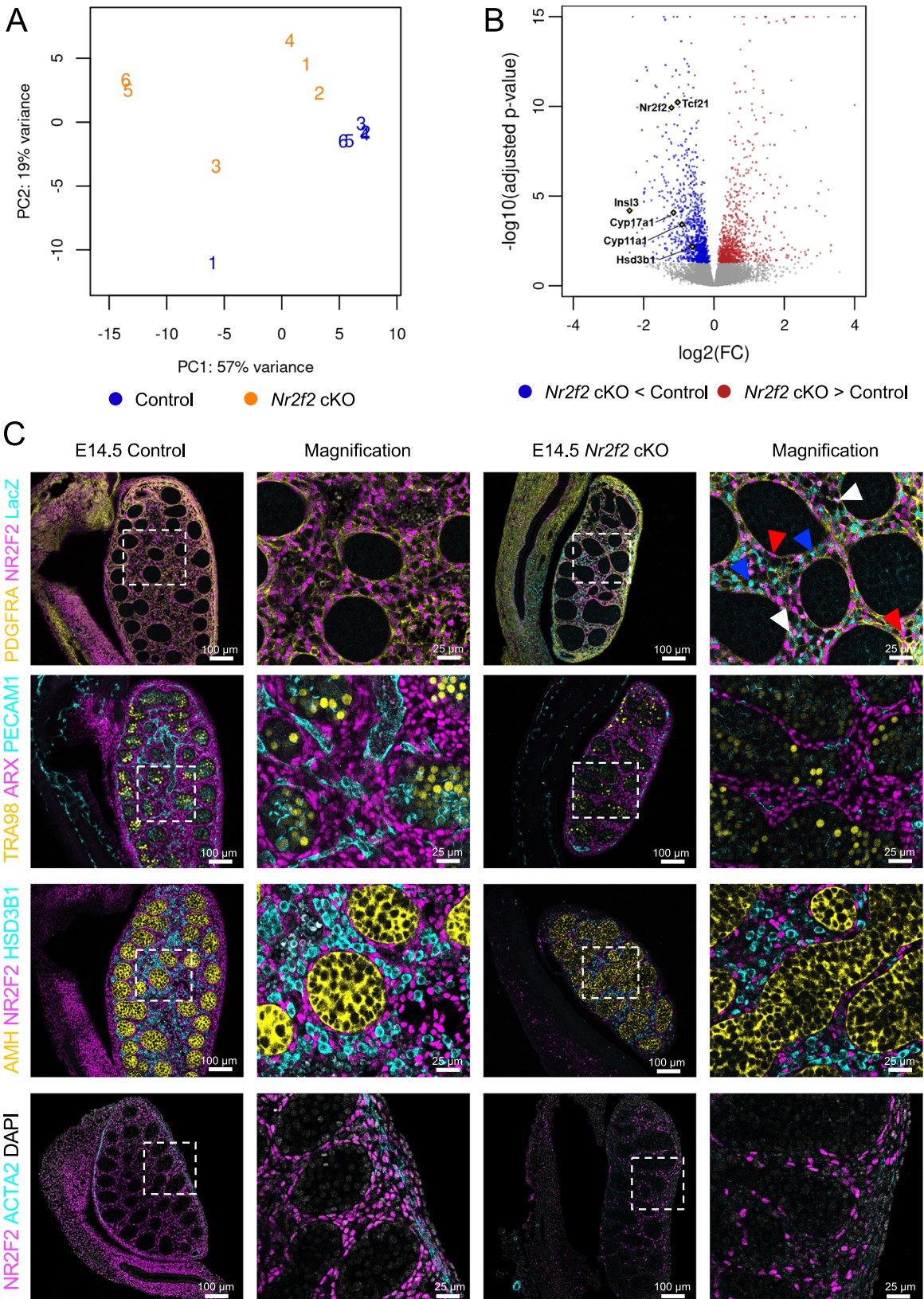

**Fig. 4 | Phenotypes of *Nr2f2* conditional knockout testes at E14.5. A** Principal component analysis (PCA) plot of the bulk RNA-seq of control and *Nr2f2* cKO testes. **B** Volcano plot of differentially expressed genes between the *Nr2f2* cKO and the control testes. Blue and red dots represent significantly (adjusted *p* < 0.05) down-regulated and upregulated genes, respectively, in the cKO testis in comparison to the control. **C** Immunofluorescence for NR2F2, LacZ and different cell type markers (PDGFRA for interstitial cells, ARX for interstitial cells, PECAM1 for endothelial cells, TRA98 for germ cells, AMH for Sertoli cell, HSD3B1 for fetal Leydig cells, and ACTA2 for contractile cells) on E14.5 control or *Nr2f2* cKO testes. White dashed box indicates the magnified area. Red arrowheads indicate NR2F2 positive/LacZ negative cells. Blue arrowheads indicate NR2F2 negative/LacZ positive cells. White arrowheads indicate NR2F2 and LacZ double positive cells.

interstitium (Fig. 4C). ARX was expressed in both NR2F2 positive/LacZ negative and NR2F2 negative/LacZ positive interstitial cells (Supplementary Fig. 4A), suggesting that NR2F2 is not required to regulate its expression, and that the phenotype observed in the *Nr2f2* knockout is not due to the absence of *Arx*[26].

Despite that ARX and PDGFRA protein expression was not altered in NR2F2-positive or negative interstitial cells, mRNA expression of these two genes was significantly downregulated in the cKO RNA-seq data. This could indicate a reduced number of interstitial cells in the cKO testes, which were smaller than their control counterparts (Fig. 4C and Supplementary Fig. 4B). Consistent with this finding, *Nr2f2* cKO testes showed a reduced interstitial to tubule seminiferous area (Supplementary Fig. 4C, D). Moreover, *Nr2f2* cKO testis exhibited a 45% reduction in total interstitial cell number per testis (Supplementary Fig. 4E), and a 25% reduction in interstitial cell density per area, accounting for the decreased testicular size (Supplementary Fig. 4F). Among the interstitial cells in *Nr2f2* cKO testis, approximately 25% were NR2F2 positive, while the remaining 75% were LacZ positive (Supplementary Fig. 4E). This corresponds to an average knockout efficiency of 75.4%, ranging between 67.3% and 86%, based on the proportion of NR2F2 and LacZ positive cells (Supplementary Fig. 4E). Collectively, these findings indicate that *Nr2f2* cKO leads to a significant reduction in non-steroidogenic interstitial cells.

E14.5 cKO testes also had a 53% reduction in HSD3B1-positive fetal Leydig cells per testicular area (Fig. 4C and Supplementary Fig. 4G). Along with downregulated expression of fetal Leydig cell markers, loss of *Nr2f2* resulted in Leydig cell hypoplasia. *Nr2f2* knockout testes also lacked SMA positive contractile cells of the tunica albuginea (Fig. 4C). This was consistent with the downregulation of smooth muscle contraction genes in the RNA-seq data (Supplementary Fig. 3C), suggesting that *Nr2f2* is also required for the differentiation or survival of these cells.

## Loss of *Nr2f2* affects interstitial, fetal Leydig cells, immune cells, and germ cells

To understand the effect of *Nr2f2* knockout in the different testicular cell populations, we performed an empirical projection of the bulk RNA-seq data into cell type specific gene expression based on the E14.5 testicular single-nucleus RNA-seq dataset (Supplementary Fig. 5). First, lists of differentially expression genes or DEGs between cKO and control testis were obtained from the bulk RNAseq results (Supplementary Fig. 5A, B). Then, relative expression patterns of these genes in the single-nucleus RNA-seq data were visualized as heatmaps (Supplementary Fig. 5), with the DEGs hierarchically grouped based on cell cluster averages (rows) and the individual cells arranged by assigned cluster (columns), thereby illustrating changes in gene expression by cell population in response to loss of *Nr2f2* (Supplementary Data 4).

Reasoning that cell populations with relatively higher expression of the DEGs are more likely to be impacted by significant changes in those genes due to loss of *Nr2f2*, we defined the "most affected" populations as those for which more than 20% of the DEGs have z-score greater than 1 across the testicular cell populations in the snRNA-seq data (Supplementary Data 4). In the case of the downregulated genes (Supplementary Fig. 5A), the predicted most "affected populations" in *Nr2f2* knockout testes were *Nr2f2*-expressing cells, including c0 (steroidogenic progenitors), c1 (contractile cells), c5 (epithelial cells) and c7 (mesonephric mesenchyme). This was consistent with the reduced number of NR2F2 positive interstitial cells observed in the cKO (Supplementary Fig. 4B–F). Gene ontology analysis of the gene groups (grp#) identified previously suggested terms like extracellular matrix organization, muscle contraction and steroid hormone biosynthesis. A key difference between *Nr2f2*-positive progenitors (c0) and *Nr2f2*-negative fetal Leydig cells (c3) is that progenitors are proliferative whereas fetal Leydig cells were non-proliferative (Supplementary Fig. 6A). Interestingly, genes related to

cell cycle progression (G2/M) (grp3) were downregulated in *Nr2f2*-expressing interstitial cells in *Nr2f2* cKO testes (Supplementary Fig. 5A). To confirm this finding, we performed immunofluorescence analysis for pH3, a marker specific for cells in late G2 and M phase (Supplementary Fig. 6B). Since LacZ expression is induced upon *Nr2f2* deletion, we quantified cell proliferation rates in both control (NR2F2 positive/LacZ negative) and *Nr2f2* knockout interstitial cells (LacZ positive/NR2F2 negative). This analysis revealed an overall decrease in interstitial cell proliferation in cKO gonads compared to controls (Supplementary Fig. 6C). Furthermore, within knockout testes, cells lacking *Nr2f2* proliferated less than those in which *Nr2f2* was not deleted (Supplementary Fig. 6D). These results indicate that *Nr2f2* is essential for interstitial cell proliferation. This defective proliferation of the progenitor cells could explain the reduction in interstitial cell number, fetal Leydig cell population, and testis size.

The empirical projection of the upregulated genes also revealed that the germ cell clusters (c4 and c6) were affected, showing upregulation of genes involved in meiotic chromosome pairing (grp2), chromosome segregation (grp2) and regulation of transcription (grp1) (Supplementary Fig. 5B), key processes attributed to male germ cells[37]. In addition, the immune cells (c8) had significant enrichment of upregulated genes in the pro-inflammatory cytokines (IL-6, IL-12 and TNF) (grp3) pathway, which could indicate immune cell infiltration and inflammation in the tissue.

## *Nr2f2* knockout results in bilateral cryptorchidism and hypospadias

Fetal Leydig cells produce hormones (androgens and insulin-like growth factor 3 or INSL3) that control testicular descent and external genitalia differentiation. We evaluated the reproductive system of *Nr2f2* cKO embryos at E17.5, right before they died due to heart conditions. The testes in *Nr2f2* cKO embryos were significantly smaller (Supplementary Fig. 7A) and located adjacent to the kidneys (Fig. 5A), in contrast to the control testes that started their descent towards the abdominal wall (Fig. 5A). This suggested a defective first phase of testicular descent, which is controlled by INSL3[41]. *Insl3* was detected as one of the most downregulated genes in the cKO testes at E14.5 (Fig. 4B). The cryptorchidism phenotype in the *Nr2f2* cKO male embryos can be explained due to a reduced expression of *Insl3* in fetal Leydig cells and/or a reduced fetal Leydig cell number. Indeed, fetal Leydig cell hypoplasia, which was observed at E14.5, remained in E17.5 cKO testes with 50% less fetal Leydig cells per area (Fig. 5B, C). Moreover, *Nr2f2* knockout testes had 70% less Leydig cells per testis than the control counterparts (Fig. 5D), which could lead to a reduction in INSL3 and androgen production.

One of the organs that require fetal Leydig cell-derived androgens is the male genitalia. Fetal Leydig cell hypoplasia and consequent low androgens during embryonic development are associated with failure of urethra closure or hypospadias in males[42–44]. Therefore, the urethra closure in control and cKO male mice was blindly examined. While E17.5 control mice had proper urethra closure on the ventral side of the penis (Fig. 5E, F), 100% of *Nr2f2* cKO male embryos had urethral defects (Fig. 5E, F). We performed immunofluorescence for the transcription factor MAFB, a critical androgen-responsive regulator of male urethra closure[45,46]. In control male mice, MAFB was located in the mesenchymal cells immediately below the urethral epithelium and in the skin (Fig. 5G). In *Nr2f2* cKO mice, MAFB-positive mesenchymal cells surrounding the urethral epithelium were drastically reduced, while the skin maintained normal MAFB expression (Fig. 5G), evidencing a defective androgen signaling.

The high penetrance of urethral defects in *Nr2f2* cKO mice suggests that fetal Leydig cells may exhibit impaired testosterone production. To evaluate their steroidogenic capacity, we performed oil red O (ORO) staining on control and *Nr2f2* cKO E17.5 testicular sections (Supplementary Fig. 7B). *Nr2f2* cKO gonads had a reduced number and

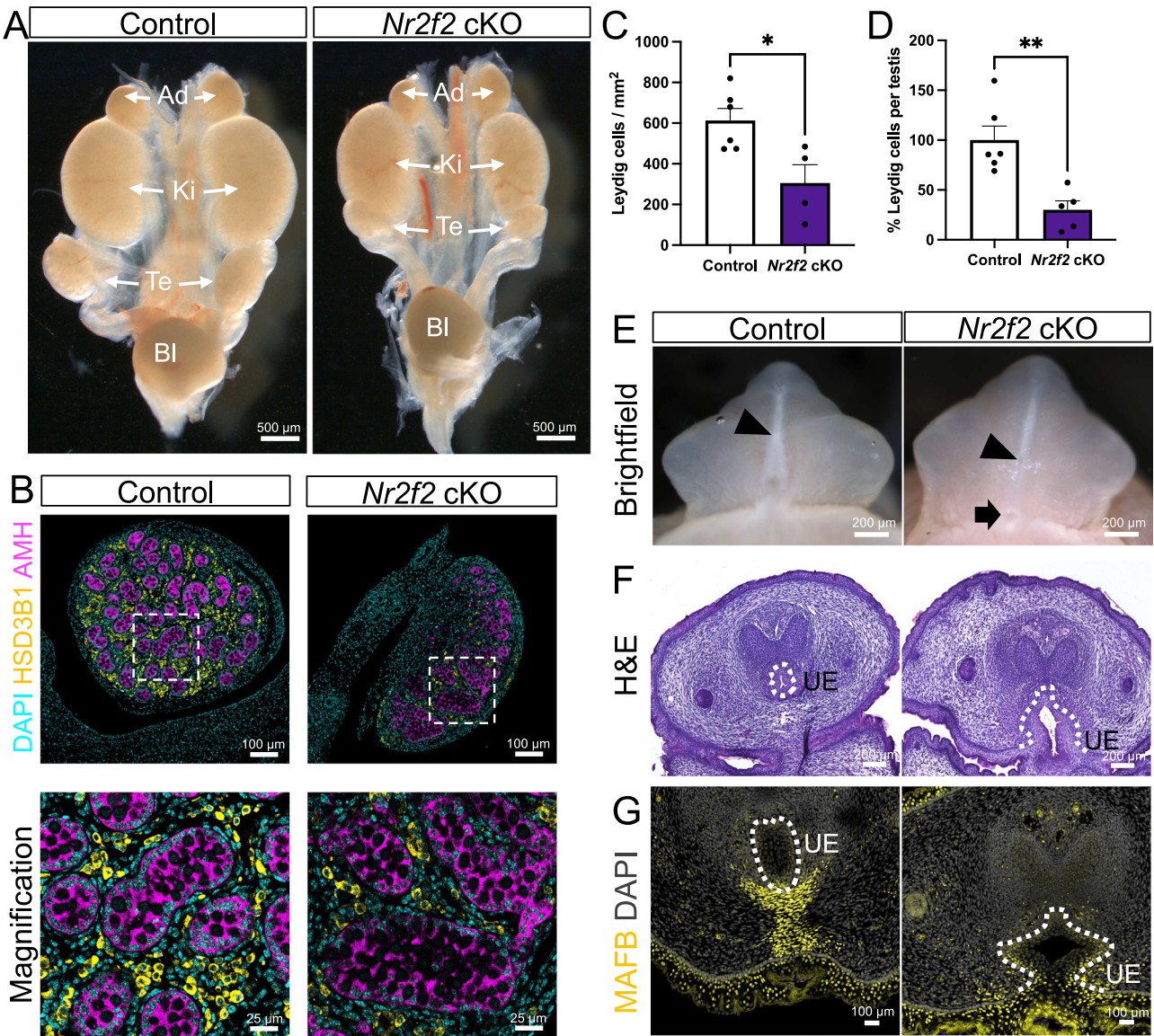

**Fig. 5 | Differences of sex development phenotypes in *Nr2f2* cKO male embryos.**
**A** Wholemount darkfield images of E17.5 urogonadal system of control and *Nr2f2* cKO male embryos. Adrenal glands (Ad), kidneys (Ki), testes (Te) and blader (Bl) are indicated with white arrows. **B** Immunofluorescence of E17.5 control or *Nr2f2* cKO testes for AMH (magenta), HSD3B1 (yellow) and DAPI (cyan). White dashed box indicates the magnified area. **C** Quantification of the number of fetal Leydig cells per area in control and *Nr2f2* cKO testes. Bars represent mean ± s.e.m., n = 6 control and n = 4 *Nr2f2* cKO, p = 0.0175. **D** Percentage of fetal Leydig cells per testis in control and *Nr2f2* cKO testes, relative to the control. Bars represent mean ± s.e.m.,

n = 6 control and n = 5 *Nr2f2* cKO, p = 0.0031. Unpaired two-tailed t-test. *P < 0.05; **P < 0.01. **E** Wholemount darkfield images of E17.5 control and *Nr2f2* cKO male genitalia. Black arrowhead represents the positioning of the urethra exit. Black arrow indicates abnormal urethra opening at the base of the genitalia.
**F** Hematoxylin and eosin staining of E17.5 control and *Nr2f2* cKO external genital sections at the base of the genitalia. **G** Immunofluorescence for MAFB (yellow) in E17.5 control and *Nr2f2* cKO genitalia section. White dashed line indicates the urethra epithelium (UE).

less intense ORO positive lipid droplets (Supplementary Fig. 7B), suggesting a defective steroidogenesis.

In summary, embryonic deletion of *Nr2f2* in mouse testes resulted in dysgenic testicles, fetal Leydig cell hypoplasia, cryptorchidism and hypospadias, which are typical features of DSDs.

### *Nr2f2* is required for tunica cells but not for PMC differentiation

Another affected cell population in the *Nr2f2* cKO testes was the contractile cell cluster (c1), which likely includes both tunica cells and peritubular myoid cells (PMCs). As PMCs are not fully differentiated by E14.5, we evaluated them at E17.5, when they are fully matured. In control testes, ACTA2, a marker of contractile cells, localized correctly to both the tunica and PMCs surrounding the testicular cords

(Supplementary Fig. 7B). However, E17.5 *Nr2f2* cKO testes lacked a defined tunica structure (Supplementary Fig. 7B), consistent with observations at E14.5 (Fig. 4C). This suggests that Nr2f2 is required for tunica cell differentiation.

Unexpectedly, SMA-positive PMCs were still present in the testis cords, in both NR2F2 positive (control) and LacZ positive (*Nr2f2* knockout) cells (Supplementary Fig. 7B). This suggests that *Nr2f2* is not necessary for PMCs differentiation. Notably, the testicular cords in the cKO testis were misshapen, suggesting that *Nr2f2* may play a role in PMCs functions, specifically in shaping the testis cord. This is consistent with downregulated pathways related to smooth muscle contraction in the cKO testis (Supplementary Fig. 3C and Supplementary Fig. 5A).

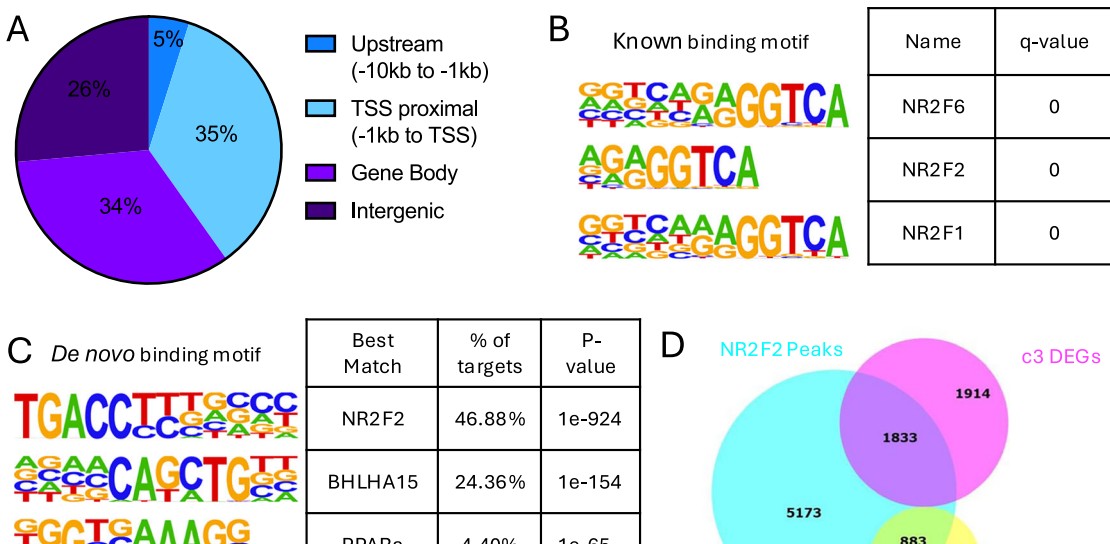

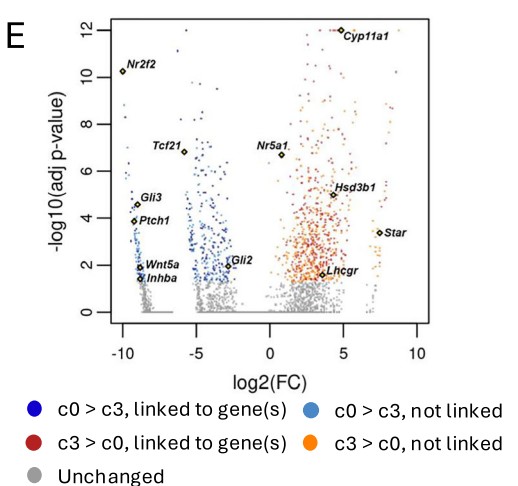

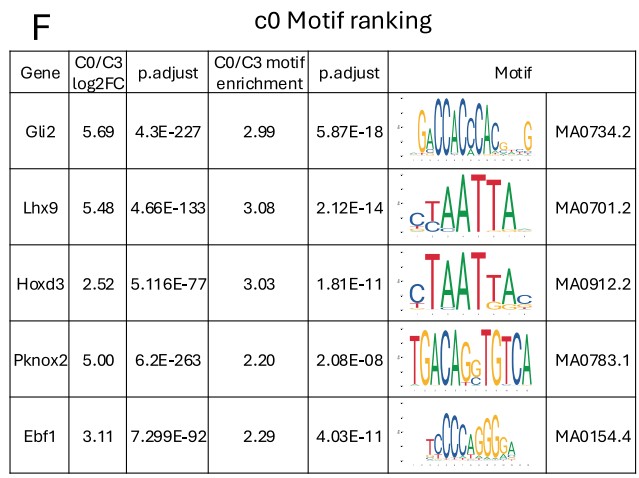

Interestingly, in the trajectory analysis, c0 and c1 are connected by a trajectory (Supplementary Fig. 2C). Moreover, pseudotime suggests that c0 contributes to c1, suggesting that c0 and c1 are developmentally related (Supplementary Fig. 2C). To assess whether Nr2f2-positive cells contribute to both the tunica and PMC lineages, we performed lineage tracing experiments using the *Nr2f2-CreER*; *CAG-*

*Sun1/sfGFP* system. *Nr2f2*+ cells were labeled at E11.5, prior to PMC and tunica cells differentiation, and analyzed for GFP with SMA co-expression at E17.5 (Supplementary Fig. 7C). GFP-positive cells were found in both SMA-positive cells in the tunica, and PMCs (Supplementary Fig. 7C), indicating that E11.5 *Nr2f2* interstitial cells give rise to these two contractile cell populations.

**Fig. 6 | NR2F2 binding and chromatin accessibility change during interstitial to Leydig cell differentiation. A** Pie chart diagram classifying the significant (fdr < $1^{-5}$) NR2F2 peaks detected based on their location in the genome. **B** Top 3 known Jaspar binding motif analysis based on the significant (fdr < $1^{-5}$) NR2F2 peaks. **C** Top 3 de novo Jaspar binding motif analysis based on the significant (fdr < $1^{-5}$) NR2F2 peaks. **D** Venn diagram contrasting all genes with a significant NR2F2 peaks (cyan) with the differentially expressed genes (DEGs) from steroidogenic progenitor cells (c0, yellow) and fetal Leydig cells (c3, magenta) based on the single-nucleus RNA-seq.

**E** Volcano plot of all the differentially accessible peaks between c0 and c3 (adjusted p < 0.05), color-coded based on the peak to gene linkage information. Expression levels and motif enrichment of the top 5 genes based on the combination of differential expression (adjusted p < 0.05) and the motif enrichment (adjusted p < 0.05) for c0 (**F**) and c3 (**G**). **H** Expression levels and motif enrichment of the top 5 genes based on the combination of differential expression (adjusted p < 0.05) and the motif enrichment (adjusted p < 0.05) for peaks containing significant (fdr < $1^{-5}$) Nr2f2 ChIP-seq peaks in c0.

## NR2F2 regulates interstitial cell identity and inhibits steroidogenesis

To understand how *Nr2f2*, which encodes a transcription factor, controls interstitial cell differentiation in fetal testes, we performed NR2F2 ChIP-seq in whole E14.5 testes (Supplementary Fig. 8). We identified 10,069 NR2F2 binding peaks (Supplementary Data 5). Among them, the biggest part (35%) were located in the proximal promoter region (−1 Kb of transcription start site or TSS), followed by the gene body (34%), intergenic (26%) and the remaining 5% was located upstream (−10 kb to 1 kb) of the most proximal gene to the peak (Fig. 6A). To confirm the specificity of the ChIP-seq antibody, we compared our data with the Homer known binding motif database. Nuclear receptor binding motifs, which share similarity in their binding sequences with NR2F2, were among the top (Fig. 6B). We also performed de novo motif analysis to generate the binding motif from our data and compare it to the existing ones. The best match for our de novo binding motif is also NR2F2 with the highest % of targets (Fig. 6C). These data provide strong support that we successfully detected the binding of NR2F2 to the testicular chromatin using ChIP-seq.

Next, we conducted pathway analyses with all NR2F2 target genes, defined as the gene most proximal to a significant NR2F2 peak (Supplementary Data 5). This analysis revealed that NR2F2 binding sites were enriched in the chromatin regions of members of the VEGFA, TGF-β, EGF and Notch signaling pathways (Supplementary Fig. 9A). VEGF and Notch signaling have been shown to inhibit fetal Leydig cell differentiation[17–19]. As NR2F2 is expressed in progenitor cells and not in fetal Leydig cells, these data suggests that NR2F2 maintains the progenitor state by stimulating both Notch and VEGF related genes. Based on the findings that most NR2F2 binding peaks were located in the TSS proximal region (-1kb to TSS), NR2F2 could directly control gene expression via the promoter region. We also performed pathway analysis using only these target genes, obtaining similar results (Supplementary Fig. 9B and Supplementary Data 5).

To further identify potential direct or indirect targets of NR2F2, we searched for differentially expressed genes (DEGs) from c0 (steroidogenic progenitors) and c3 (fetal Leydig cells) from our single-nucleus analysis that contain NR2F2 peaks from the ChIP-seq analyses (Fig. 6D and Supplementary Data 5). Around 50% of DEGs in either c0 (883 out of 1,648 DEGs) or c3 (1,833 out of 3,747 DEGs) had NR2F2 binding peaks (Fig. 6D). Pathway analyses of the DEGs in the c0 cluster with NR2F2 peaks revealed enrichments of the extracellular matrix organization (Supplementary Fig. 9C and Supplementary Data 5). Surprisingly, in interstitial cells, NR2F2 binding peaks were detected in regulatory regions of genes associated with fetal Leydig cells (c3), such as *Nr5a1* and *Star* (c3), some of them involved in lipid metabolism and cholesterol biosynthesis (Supplementary Fig. 9D, Supplementary Data 5). These results suggest that in steroidogenic progenitor, where NR2F2 is expressed, genes associated with mesenchymal cell lineages were stimulated, while NR2F2 represses the expression of Leydig cell-specific genes in interstitial cells. During progenitor-to-Leydig cell differentiation, NR2F2 is downregulated in fetal Leydig cells, therefore allowing the de-repression and consequent expression of steroidogenic genes. Intriguingly, steroidogenic genes were not upregulated in *Nr2f2* cKO testis, implicating the potential involvement of other unknown inducers of fetal Leydig cells.

## Identification of candidate transcription factors regulating interstitial to fetal Leydig cell differentiation

Our single-nucleus multiomic data revealed that although the steroidogenic progenitors (c0) and fetal Leydig cells (c3) transcriptome varied substantially (5395 DEGs) (Fig. 1A), their chromatin accessibility changes were less abrupt (Fig. 1E). A total of 1070 regions were differentially accessible between c0 and c3, and among them 386 were more accessible in c0 than c3, and 684 more accessible in c3 than c0 (Fig. 6E, Supplementary Data 6). We then performed peaks to genes correlation analysis, which identify chromatin accessible peaks (single-nucleus ATAC-seq) correlated with gene expression (single-nucleus RNA-seq) on each sequenced nucleus. Around 48.7% (188/386) and 47.9% (328/684) of the differentially accessible peaks were linked to genes in c0 and c3, respectively (Fig. 6E, Supplementary Data 6). Pathway analysis of these linked genes consistently identified pathways related to c0 steroidogenic progenitors cluster (extracellular matrix organization, hedgehog/GLI and TGFB signaling; Supplementary Fig. 9E and Supplementary Data 6) and c3 fetal Leydig cell cluster (metabolism of steroid, lipids and steroid hormones; Supplementary Fig. 9F and Supplementary Data 6).

Less pronounced changes in chromatin accessibility compared to the transcriptome between c0 progenitors and c3 fetal Leydig cells (Fig. 1) suggests that interstitial to Leydig cell differentiation is primarily driven by transcription factors rather than extensive chromatin remodeling. To identify the transcription factors involved in interstitial to Leydig cell differentiation, we performed motif analyses for potential transcription factor binding sites in the differentially accessible chromatin regions between c0 and c3 (Supplementary Data 7). We focused only on transcription factors that are differentially expressed between c0 and c3 based on single-nucleus RNA-seq data. Such analysis yielded a total of 52 and 17 transcription factors for c0 and c3, respectively (Supplementary Data 7). To identify the top potential transcription factors binding to the differentially accessible chromatin on each cell cluster, we ranked these transcription factors based on the combination of their differential mRNA expression and their motif enrichment score (Supplementary Data 7). For the c0 steroidogenic progenitor cluster, previously reported transcription factors were ranked high in the steroidogenic progenitor cells, including *Gli2*, *Lhx9*, *Tcf21*, *Wt1* and *Arx* (Supplementary Data 7). The other top candidates in the c0 cluster included transcription factors *Hoxd3*, *Pknox2* and *Ebf1* (Fig. 6F). For the c3 fetal Leydig cell cluster, the top 5 transcription factors were *Nr5a1*, *Esrrg*, *Rora* and *Thrb*, and *Gata6* (Fig. 6G).

We next investigated the association between accessible chromatin regions and NR2F2 binding enrichment during interstitial to Leydig cell differentiation. We combined the single-nucleus ATAC-seq data with our NR2F2 ChIP-seq data (Supplementary Data 6). Roughly 43.7% (169) of the differentially accessible peaks in the progenitor cells (c0) contained a significant NR2F2 binding peak (Supplementary Data 6). On the other hand, only 3.8% (26) of the differentially accessible peaks in the fetal Leydig cell cluster (c3) contained a significant NR2F2 binding peak (Supplementary Data 6). Pathway analysis of the genes linked to accessible chromatin peaks containing an NR2F2 significant binding peak in c0 suggest an involvement of NR2F2 in regulating collagen formation and hedgehog/GLI pathway

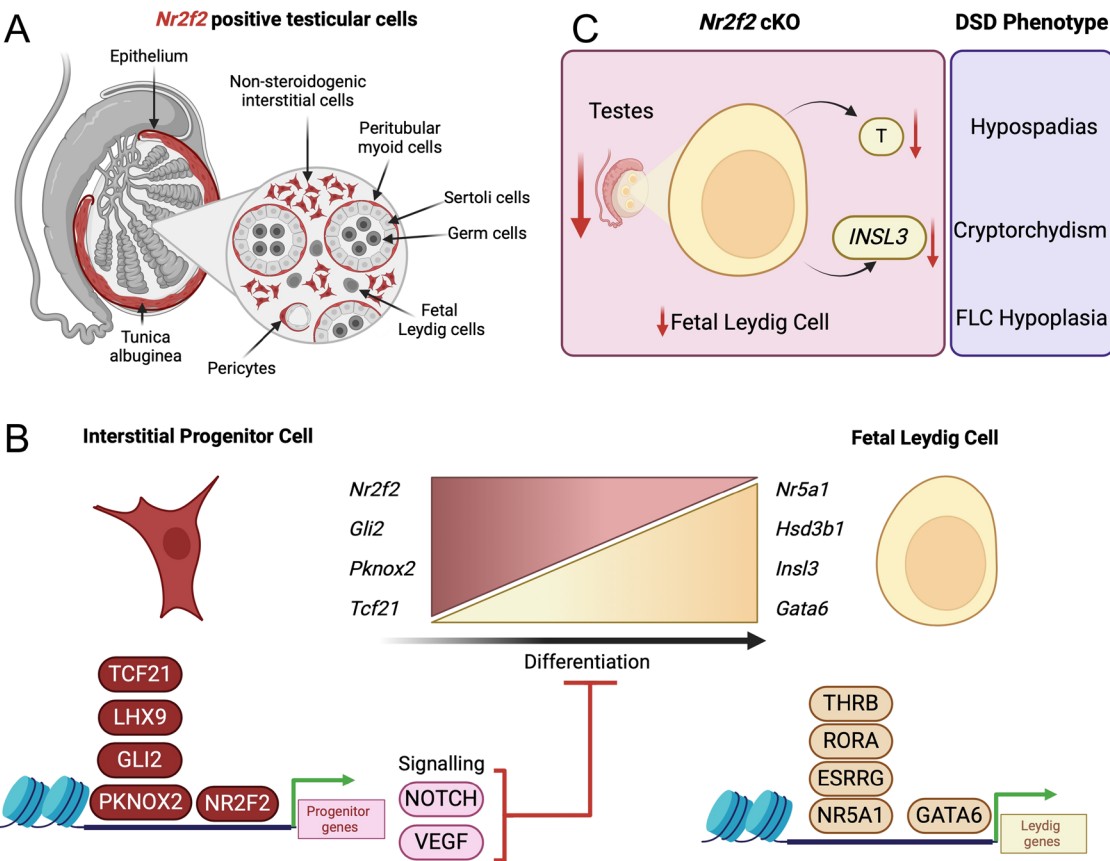

**Fig. 7 | Role of NR2F2 in testicular morphogenesis. A** Schematic representation of the different *Nr2f2*-positive cell populations identified with our single-nucleus multiomic analysis. **B** Schematic representation of how NR2F2 controls the identity of the interstitial cells while repressing Fetal Leydig cell differentiation. **C** Summary figure of the DSD phenotypes in *Nr2f2* conditional knockout testis. Created in BioRender, Yao, H. (2025) https://BioRender.com/zg96fxw.

(Supplementary Fig. 9G, Supplementary Data 6). Surprisingly, despite 43.7% of the differentially accessible peaks in the progenitor cells (c0) containing a significant NR2F2 binding peak, NR2F2 motif was not enriched in the c0 motif analysis. One possible explanation was that NR2F2 regulates the expression of other transcription factors in the c0 clusters. Indeed, 73% (38/52) of the different transcription factors identified in the c0 motif analysis had a NR2F2 ChIP-seq binding peak (Supplementary Data 7). Moreover, 8 of them were downregulated in *Nr2f2* knockout testes, including *Pknox2* and *Tcf21* (Supplementary Data 7). This suggests that NR2F2 acts upstream of a cohort of transcription factors such as *Pknox2* and *Tcf21* to regulate their expression and assuring interstitial cell identity. Another possible explanation is that these factors act as co-regulators with NR2F2 to control the expression of progenitor-specific genes. To identify potential co-regulators of NR2F2, we performed motif analysis on c0 differentially accessible peaks that overlapped with NR2F2 ChIP-seq binding sites (169 peaks) (Supplementary Data 7). This analysis identified 49 potential transcription factors expressed in c0 steroidogenic progenitor cells that bind to the same chromatin regions as NR2F2 (Supplementary Data 7). Among the top ranked factors, we found the same transcription factors, such as *Gli2*, *Pknox2*, *Lhx9*, *Ebf1*, *Smad3* and *Hoxd3*, but in a different order (Fig. 6H, Supplementary Data 7). There was an enrichment of *Pknox2* and *Smad3* binding motifs specifically in open chromatin regions containing NR2F2 ChIP-seq peaks in c0 steroidogenic progenitors.

## Discussion

Using single-nucleus multiomics analysis and lineage-tracing mouse models, we characterized the cellular landscape of the embryonic testis with a particular focus on interstitial cell populations (Fig. 7A). Our study revealed that *Nr2f2*-positive cells serve as progenitors for contractile cells and fetal Leydig cells and highlighted the critical role of NR2F2 in establishing interstitial cell identity, fetal Leydig cell differentiation, and overall testicular morphogenesis (Fig. 7B). Embryonic deletion of *Nr2f2* in mouse testes led to phenotypes of DSDs, including dysgenic testicles, fetal Leydig cell hypoplasia, cryptorchidism, and hypospadias (Fig. 7C). These phenotypes mirror human DSDs associated with *NR2F2* mutations[32]. A similar phenotype was observed by knocking out *Nr2f2* in *Nr5a1* positive cells or in *Wt1* positive cells at earlier timepoints (E9.5 and E10.5), confirming the important role of *Nr2f2* in fetal Leydig cell differentiation[47]. Additionally, through an integrated analysis of single-nucleus RNA-seq, ATAC-seq, and NR2F2 ChIP-seq data, we identified NR2F2 as a master regulator of interstitial cells. While promoting the interstitial cell fate, NR2F2 inhibits fetal Leydig cell differentiation by modulating expression of components in VEGF and Notch signaling pathways at the promoter level and controls the expression of genes involved in extracellular matrix formation and key chromatin remodelers such as *Pknox2* and *Tcf21* (Fig. 7B).

Fetal Leydig cells play an important role in male sexual differentiation. Despite being described since the 1900's[6,7], the origin of Leydig cells and in particular fetal Leydig cells has been heavily debated. Existing data suggest a dual origin of fetal Leydig cells, either differentiating from the coelomic epithelium through epithelial to mesenchymal transition[48,49], or migrating into the gonad from the adjacent mesonephros[8,50,51]. Using our single-nucleus multiome data, we identified the *Nr2f2*+ interstitial cell population (c0) as the progenitors of fetal Leydig cells. The mesonephric interstitial cells (c7), on the other hand, were not identified as Leydig cell progenitors.

Nevertheless, we cannot exclude the mesonephric contribution to the fetal Leydig lineage before or after E14.5. The *Nr2f2* lineage tracing results further support that *Nr2f2*-positive cells are the progenitors of fetal Leydig cells. While differentiating into fetal Leydig cells, *Nr2f2*-positive progenitor cells downregulate NR2F2 and upregulate steroidogenic genes such as NR5A1, HSD3B1 and CYP17A1 (Fig. 7B). Previous single-cell RNA-seq and *Wnt5a* lineage tracing experiments indicated that *Wnt5a* positive steroidogenic progenitors cells are specified between E11.5 and E12.5, giving rise to the majority of fetal and adult Leydig cells[10]. Unfortunately, a detailed lineage tracing involving previous (E10.5-E11.5) or later timepoints (E12.5-E13.5) was not provided to confirm that the progenitors are specified only during this short period of time[10]. The origin of the steroidogenic progenitor cells and the exact time of their specification remains to be determined, which requires lineage tracing experiments before E11.5. As *Nr2f2* deletion in Wt1 positive cells at E9.5 and E10.5 also resulted in fetal Leydig cell deficiency, we hypothesize that *Wt1* positive cells, at E9.5 and/or E10.5, are either positive for *Nr2f2* or they are the progenitors of the *Nr2f2*+ steroidogenic progenitor cells[47]. Based on our lineage tracing experiments at E11.5, E12.5, E13.5, and E14.5, we found that *Nr2f2* positive cells at all the evaluated timepoints act as Leydig progenitor cells, suggesting that ability of the progenitor cells to differentiate into fetal Leydig cells expand beyond E11.5 and E12.5. We therefore propose that progenitor-to-fetal Leydig cell differentiation is a continuous process, occurring, at least, between E11.5 and E14.5, from a pool of Nr2f2-positive progenitor cells in the interstitium.

One of the first markers described for Leydig cell progenitors was the transcription factor *Arx*[26]. ARX and NR2F2 colocalize in the testis interstitium, and both are downregulated during interstitial to Leydig cell differentiation. Phenotypes of global *Arx* deletion mirrored our *Nr2f2* cKO phenotypes (dysgenic, undescended testes and fetal Leydig cells hypoplasia), suggesting a similar role of ARX in interstitial to Leydig cell differentiation[26]. Unfortunately, no lineage tracing results are available to confirm that *Arx*-positive cells are the fetal Leydig cell progenitors[26]. In *Nr2f2* cKO testis, ARX continues to be expressed in LacZ positive/NR2F2 negative interstitial, suggesting that NR2F2 does not regulate *Arx* gene expression. It is unknown if ARX controls *Nr2f2*. Other than *Nr2f2* and *Arx*, *Tcf21*[25] (or *Pod1*) and *Wnt5a*[10] also labeled fetal Leydig cell progenitors in lineage tracing experiments. Their potential role during this differentiation process has not been delineated.

Our single-nucleus multiomic dataset of E14.5 testes revealed more cellular complexity than expected. Among the previously described clusters, including Sertoli cells, fetal Leydig cells, supporting-like cells, and germ cells, we identified two different endothelial cell populations and one immune population. These populations will require further investigation to understand their role in testicular development and function. In addition, we identified and characterized 4 interstitial cell clusters and an epithelial cluster with relatively similar transcriptomic expression and chromatin accessibility. These clusters could be developmentally related. As NR2F2 is not only expressed in the non-steroidogenic interstitial cells, but also in the contractile cells, epithelium and mesonephros, we cannot rule out the possibility of a secondary origin of the steroidogenic fetal Leydig cells from these clusters. We also identified a pericyte cell cluster (c10) based on the expression of the pericyte markers *Mcam*, *Steap4* and *Cspg4* (NG2)[18,52]. In adult humans and macaque, two different pericyte populations have been identified, one muscular, expressing *MCAM* and the other fibroblastic, expressing *STEAP4*[52]. Our mouse dataset only identified one pericyte population, co-expressing both fibroblastic and muscular genes. One hypothesis is that mouse testis has only one type of pericytes, unlike humans and macaques. Another possibility is that this pericyte population is the embryonic progenitor of the muscular and fibroblastic pericytes in the adult. Further research is required to elucidate the developmental origin of this population.

In addition to fetal Leydig cells, another severely affected population in the *Nr2f2* cKO testes was the contractile cells, particularly the ones in the tunica albuginea. The tunica albuginea, also known as testicular capsule, is a contractile muscle layer that surrounds the testis. Due to its rhythmical contractions, it is theorized to have a major role in sperm transportation towards the rete testis and epididymis[53,54]. We could not examine such functions in the *Nr2f2* cKO mice due to their perinatal lethality. Further research targeting only the tunica cells will be key to determine the exact role of the tunica in testicular morphology, development and function. The trajectory and pseudo time analysis suggested that the *Nr2f2*+ interstitial cell population (c0) serves as a progenitor of the contractile cells (c1). This differentiation process was also confirmed by lineage tracing experiment of *Nr2f2*+ cells at E11.5, before the emergence of contractile cells. As *Nr2f2* is expressed in tunica cells, we cannot conclude if the lack of tunica cells in the *Nr2f2* cKO is due to lower number of progenitor cells, or if *Nr2f2* is required to trigger tunica cell differentiation. Future research using c0 and c1 specific markers is required to evaluate the roles and contribution of these two cell populations to testicular differentiation and function.

By combining single-nucleus multiomics, ChIP-seq and bulk RNA-seq data of *Nr2f2* cKO testes, we interrogated the potential mechanism of NR2F2 action. NR2F2 promotes interstitial identity by stimulating genes associated with mesenchymal properties such as extracellular matrix genes. On the other hand, NR2F2 inhibits fetal Leydig cell differentiation by inducing the expression of components in the VEGF and Notch signaling pathways at the promoter level (Fig. 7B). VEGF-VEGFR signaling pathway represses Leydig cell differentiation, as inhibition of VEGFR kinase activity with the VEGFR Tyrosine Kinase Inhibitor II led to an expansion of fetal Leydig cell[19]. Similarly, Notch signaling pathway also inhibits Leydig cell differentiation, as constitutive activation of the Notch pathway suppressed Leydig cell differentiation[17]. Conversely, when Notch signaling was inhibited, fetal Leydig cell hyperplasia occurred[17,18]. NR2F2 binds to the regulatory regions of genes involved in cholesterol and lipid metabolism, which are enriched in fetal Leydig cells. This suggests that NR2F2 could inhibit the expression of steroidogenic genes, blocking Leydig differentiation and maintaining the progenitor status (Fig. 7B).

This evidence leads to our hypothesis that as progenitor cells differentiate into fetal Leydig cells, *Nr2f2* expression is downregulated, therefore allowing the derepression of steroidogenic genes for androgen production. If indeed this is the case, one would expect that in the absence of NR2F2, interstitial cells would differentiate into fetal Leydig cells. In fact, fetal Leydig cell number was reduced in the *Nr2f2* cKO testis, implying that loss of NR2F2 and removal of its repressive action were not sufficient to induce fetal Leydig cell differentiation. We speculate that other positive transcriptional regulators are required to facilitate interstitial to Leydig cell differentiation. To identify such factors, we performed motif analysis on differentially accessible chromatin regions obtained from the single-nucleus multiomic datasets. We identified 17 potential transcription factors in the fetal Leydig cells, including the nuclear receptors *Nr5a1*, *Esrrg*, *Thrb* and *Rora*, which shared similar binding motifs, and *Gata6* (Fig. 7B). *Nr5a1* and *Esrrg* are known to be important for Leydig cell differentiation and steroidogenesis[55,56], whereas the role of *Rora* and *Thrb* is unknown. Although *Gata6* is expressed in Leydig cells and it has been shown that Gata6 and Gata4 double knockout results in smaller testes with fewer fetal Leydig cells, the specific contribution of *Gata6* to this phenotype needs to be elucidated[57,58]. Despite not being expressed in fetal Leydig cells, NR2F2 binding motifs were enriched in the fetal Leydig cell motif analysis. This could suggest that in interstitial cells, NR2F2 binds to Leydig cell associated genes and inhibits the differentiation towards fetal Leydig cells, as suggested by our ChIP-seq data. During fetal Leydig cell differentiation, *NR2F2* is downregulated, therefore de-occupying these open chromatin sites and becoming accessible to

other Leydig cell transcription factors like *Nr5a1*, *Esrrg*, *Thrb* and *Rora* (Fig. 7B). Interestingly, while *Nr2f2* is not expressed in non-proliferative mouse fetal Leydig cells, *Nr2f2* is expressed in proliferative mouse Leydig cell tumor cell lines, such as MA-10 and MLTC-1, where it cooperates with *Nr5a1* to drive the expression of steroidogenic genes[52,59,60]. As testicular *Nr2f2* knockout impacted the proliferative state of interstitial cells, *Nr2f2* misexpression in Leydig cell may contribute to tumorigenesis by promoting cell proliferation. A similar role for *Nr2f2* has been observed in malignant squamous carcinoma, where NR2F2 enhanced tumor cell proliferation and stemness, while suppressing differentiation[61]. Further research is needed to clarify the role of *Nr2f2* in testicular cancer.

Another possible explanation for the reduction in fetal Leydig cells in the *Nr2f2* cKO testis is that, in the absence of *Nr2f2*, interstitial cells lose the ability to properly differentiate into fetal Leydig cells. *Tmsb10* is known to be expressed in steroidogenic progenitor cells and required to initiate Leydig cell differentiation[30], and our single cell analysis found *Tmsb10* is enriched in progenitor cells, but not fetal Leydig cells. A significant NR2F2 binding peak was identified at the TSS of *Tmsb10* gene (Supplementary Data 5), suggesting a direct regulation by NR2F2. Additionally, *Tmsb10* was downregulated in *Nr2f2* cKO testis. These findings together suggest that NR2F2 directly regulates *Tmsb10* expression. Given the importance of *Tmsb10* in the differentiation of interstitial cells into fetal Leydig cells, the reduced *Tmsb10* expression observed in *Nr2f2* cKO testes could contribute to not only a reduction in interstitial cells, but also impaired capacity for progenitor cells to properly differentiate into fetal Leydig cells. The presence of oil droplets in the remaining fetal Leydig cells in our *Nr2f2* knockout model implied that these fetal Leydig cells were capability of producing steroids. However, whether they are functionally normal cannot be discerned.

Our analysis also identified key transcription factors that could be involved in regulating interstitial cell identity, including previously reported *Gli2*, *Tcf21* and *Arx*[25,26,29,62] (Fig. 7B). *Tcf21* global knockout mice have hypoplastic testes with upregulation of steroidogenic genes, including *Sf1* and *Cyp11a1* (scc)[29]. Such observation suggests that *Tcf21* not only promotes the interstitial cell fate but also represses Leydig cell differentiation. Surprisingly, despite *Tcf21* being downregulated in our *Nr2f2* cKO testes, steroidogenic genes were not upregulated, but rather downregulated. This suggests that *Tcf21* downregulation is not sufficient to drive fetal Leydig cell differentiation in the absence of *Nr2f2*. As our ChIP-seq data revealed that *Tcf21* is a target of NR2F2, and NR2F2 also binds and inhibits steroidogenic gene expression, it would be interesting to explore the coregulation of these factors during interstitial to Leydig cell differentiation.

Our Motif analysis also identified transcription factors such as HOXD3, PKNOX2 and EBF1 as potential regulators of the interstitial cell chromatin identity (Fig. 7B). Although *Pknox2*, *Hoxd3*, and *Ebf1* have not been previously associated with testicular development, they are expressed in cardiac, lung and skin fibroblasts, other sites of *Nr2f2* expression[63-65]. These factors could represent a potential fibroblastic signature co-regulating with NR2F2 the interstitial cell identity, making them interesting candidates for further characterization.

Variants in the *Nr2f2* gene are associated with XY and XX DSDs in humans[32,66,67]. A de-novo missense variant was identified in a 46, XY individual with micropenis, cryptorchidism and hypospadias[32]. Our *Nr2f2* knockout mouse model recapitulated most of the human phenotypes, including dysgenic testes, reduced testicular interstitium, Leydig cell hypoplasia, cryptorchidism and hypospadias (Fig. 7C). The *Nr2f2* cKO testes were not able to produce sufficient testosterone and INSL3 to facilitate proper genital differentiation and testicular descent, resulting in hypospadias and cryptorchidism. This research identifies molecular targets that could enhance the diagnosis and management of *Nr2f2* associated DSDs. Taken together our findings underscore the significant role of *Nr2f2* in regulating testicular development, morphogenesis and function.

## Methods

### Animals

*Nr5a1-cre* mice (*B6D2-Tg(Nr5a1-cre)2Klp*) were provided by the late Dr. Keith Parker[68]. *Nr2f2f/f* (*B6;129S7-Nr2f2<tm2Tsa >/Mmmh*) mice were provided by Dr. Sofia Tsai[40]. *Nr2f2-CreER* mice (*B6.129S-Nr2f2<tm1(icre/ERT2)Hy*) were generated by NIEHS gene editing and mouse model core facility[39] *Rosa-tdTomato* (*B6.Cg-Gt(ROSA)26Sor<tm9(CAG-tdTomato)Hze >/J*), *CAG-Sun1/sfGFP* (*B6;129-Gt(ROSA)26Sor<tm5(CAG-Sun1/sfGFP)Nat >/J*) and *Wt1-CreER* (*Wt1<tm2(cre/ERT2)Wtp >/J*) mice were purchased from Jackson Laboratory (stock number 007909, 021039 and 010912, respectively). Mice were housed on a 12 h light:dark cycle, temperature range 21–23 °C, and relative humidity range from 40 to 50%. Pairs were set up overnight and females were screened for the presence of vaginal plug the next morning. Positive detection of vaginal plug was considered embryonic day (E) 0.5. All animal procedures were approved by the National Institute of Environmental Health Sciences (NIEHS) Animal Care and Use Committee and are in compliance with a NIEHS-approved animal study proposal (2010-0016).

### Single-nucleus multiomics

Whole testes were collected from E14.5 *Nr5a1-cre; Rosa-tdTomato* embryos in PBS and immediately frozen in liquid nitrogen for storage. Nuclei isolation was performed using the 10X Genomics Chromium Nuclei Isolation Kit (1000494) following the manufacturer's protocol. The 10X Genomics Chromium Next GEM Single Cell Multiome ATAC + Gene Expression Library Preparation Kit (1000284) was used to generate a 10X barcoded library of mRNA and transposed DNA from individual nucleus. The 10X barcoded single-nucleus RNA and ATAC-seq libraries were sequenced by the NIEHS Epigenomics and DNA Sequencing Core on an Illumina NovaSeq 6000 instrument.

### Multiomics analysis and cluster annotation

The single-nucleus multiome (snMultiome) data was initially processed by the 10x Genomics Cell Ranger ARC v2.0.1[69,70] count function at default settings with gene model version arc-mm10-2020-A-2.0.0. Putative doublets were identified with scDblFinder v1.16.0[71], which was run independently on the RNA assay (at default parameters) and the ATAC assay (with parameters aggregateFeatures=TRUE, nfeatures=25, processing = "normFeatures"); cells predicted to be doublets by both assays were removed from the dataset. Ambient RNA decontamination was performed with the DecontX function of the celda package v1.18.1[72]. Seurat v5.0.1[73] and Signac v1.12.0[74] were used to evaluate data quality; cells were filtered based on both the RNA assay (required <10% mitochondrial gene content, <10% ribosomal gene content, <0.5% hemoglobin gene content, and >500 detected genes per cell) and the ATAC assay (required >20% of reads at peaks, >250 but <25,000 fragments at peaks, and <5% of fragments at blacklist regions) (Supplementary Fig. 10). The RNA and ATAC assays were first processed separately to allow visualization of the cells by UMAP specific to each data type, then subsequently processed together via the Seurat FindMultiModalNeighbors function. Clusters were assigned according to the results of Seurat FindClusters at resolution 0.3. Genes differentially expressed between specific clusters were identified by the Seurat FindMarkers function with adjusted p-value threshold set at 0.05. Single-cell visualizations were generated by the Seurat DimPlot, FeaturePlot, and DotPlot functions. Cell cycle phase summary per cluster was based on phase per cell reported by the Seurat CellCycleScoring function. Monocle3 was used to generate cell trajectories based on the snATAC-seq data[75]. The Euclidean distance between each pair of clusters was calculated based on averages of the per-cell snATAC-seq signal at each of the 2000 most variable snATAC-seq peaks (as identified with the FindVariableFeatures function in Seurat); the hclust R

function was then used to generate a cluster dendrogram from the resulting distance matrix.

To focus on cell populations of interest, snATAC-seq peaks were subsequently called by MACS2 v2.2.9.1[76] based only on signal from cells in clusters c0 and c3, then filtered to retain only peaks on canonical chromosomes (chr1-19,X,Y). Predicted links between snATAC-seq peaks and genes were reported by the Signac LinkPeaks function. Differentially accessible peaks were identified by the Seurat FindMarkers function using parameters test.use = "LR" and latent.vars = "peak_region_fragments", with adjusted p-value threshold set at 0.05. Motifs enriched in differentially accessible peaks were identified by the Signac FindMotifs function, from a collection of JASPAR2020[77] motifs specified using TFBSTools[78] function getMatrixSet with options collection = "CORE", tax_group = "vertebrates", and all_versions=FALSE. All snMultiome analyses described above were performed in R v4.3.2.

Clusters were annotated using marker genes known for different cell types[1,10] (Supplementary Data 1). Sertoli cells (cluster 2) express *Sox9*, *Amh* and *Dmrt1*. Cluster 3 was labeled as fetal Leydig (*Nr5a1*, *Cyp17a1* and *Hsd3b1*). Cluster 9 was classified as *Pax8*[+] supporting-like cells[1]. Clusters 11 and 12 were classified as endothelial (*Pecam1* positive). Cluster 8 was assigned as immune cells/macrophages (*Adgre1* positive). Clusters 4 and 6 were classified as *Ddx4* and *Dazl* positive germ cells. E12.5 and E13.5 multiomics data was analyzed as previously described[36,37].

## Lineage tracing
Heterozygous Nr2f2-CreER or Wt1-CreER mice were crossed with homozygous CAG-Sun1/sfGFP mice. CreERT2 activity was induced by intraperitoneal injection of 1 mg of tamoxifen (T-5648, Sigma-Aldrich) in corn oil per 10 g of mouse body weight. For *Nr2f2-CreER* model, tamoxifen was injected at E11.5, E12.5, E13.5 or E14.5. For *Wt1-CreER* model, tamoxifen was injected at E11.5 and E12.5. Samples were collected at E14.5 or E17.5 and processed for immunofluorescence analysis.

## Immunofluorescence and histological analyses
Fetal testes and genitalia were collected and fixed in 4% PFA overnight at 4 °C. Gonads and genitalia were then washed in PBS and stored at 4 °C in 70% ethanol and posteriorly paraffin embedded. Five μm paraffin sections were dewaxed, rehydrated and subjected to antigen retrieval using a pre-heated (5 min at 100% microwave power) 1/1000 dilution of citrate-based antigen unmasking solution (Vector Labs, H-3300-250) for 20 min in the microwave at 10% power. Samples were cooled at room temperature for 30 min, then transferred to the Sequenza manual immunohistochemistry system in PBS using 100 μl of solutions, unless specified. Samples were blocked and permeabilized in 0.1% triton X-100 in 1X PBS with 5% normal donkey serum (blocking buffer) for 1 h. Samples were incubated with primary antibodies (Supplementary Data 8) diluted in blocking buffer at 4 °C overnight. Samples were then washed three times with 0.1% triton X-100 in 1X PBS and incubated with secondary antibodies (Supplementary Data 8) diluted in blocking buffer at room temperature for 1 h. Slides were washed one time with 0.1% triton X-100 in 1X PBS, followed by two times with 1X PBS and and incubated with the TrueView Autofluorescence Quenching Kit (Vector Labs, SP-8400) for 3 min. Samples were washed with 500 μl of 1X PBS, counterstained with DAPI (Invitrogen, D1306), washed one more time in 1X PBS and mounted using ProLong™ Diamond Antifade Mountant (Invitrogen, P36970). Imaging was performed on a Zeiss LSM 900 confocal microscope using Zen software. Brightness and contrast of images were adjusted using FIJI[79].

To determine the histological structure of the external genitalia, we used hematoxylin and eosin staining. Briefly, slides were deparaffinized with xylene and rehydrated through an ethanol gradient. Slides were then stained with Harris' hematoxylin, rinsed with tap water and differentiated with 0.3% acid alcohol (Ethanol – HCl). After rinsing with tap water and bluing solution, slides were stained with eosin. Slides were then dehydrated with ethanol, cleared with xylene, and cover slips were mounted with Permount. Sections were imaged under the Keyence Bz-X810 microscope (Keyence, Osaka, Japan) with a 20x Objective lens magnification.

## Oil Red O staining
E17.5 testes were snap frozen in dry ice and stored at −80 °C until utilization. Samples were fixed in 4% PFA for 1 h at room temperature, washed 3 times for 10 min with 1X PBS and cryo-protected in 30% sucrose overnight. Samples were embedded in OCT and 10 μm sections were cut using a cryostat. Oil Red O staining was as described before[80]. Briefly, slides were circled with a hydrophobic pen and 0.5 ml of ORO working solution was added to each slide. Samples were incubated for 10 min at room temperature and washed in running water for 30 min. Samples were then incubated with DAPI in 1X PBS for 10 min at room temperature in the dark and washed in water. Samples were mounted using aqueous antifade mounting medium (Vectashield, H-1000-10) and coverslips were sealed with nail polish. Taking advantage of the far-red Oil Red O fluorescence, slides were imaged a couple of hours after the samples were mounted on a Zeiss LSM 900 confocal microscope using Zen software.

## Statistics and reproducibility
All histological stainings were performed on embryonic tissues from mouse embryos collected from at least three independent litters. Embryos from independent litters were analyzed to ensure reproducibility. Representative images are shown. Gonadal area, testicular cord area and cell number were quantified on gonadal paraffin sections using FIJI[79]. Interstitial area was defined as gonadal area minus testicular cord area. Data was visualized and statistical analysis (unpaired or paired two-tailed t-test) was performed using GraphPad Prism (*P < 0.05; **P < 0.01;***P < 0.001; ****P < 0.0001).

## Chromatin immunoprecipitation sequencing or ChIP-seq for NR2F2
Testes from E14.5 CD-1 (Charles River stock number 022) were separated from the mesonephros, snap-frozen, and stored at −80 °C. Two independent ChIP-seq experiments were performed by Active Motif, Inc., using 18 μg of sheared chromatin from 30 pooled embryonic testes and 5 μl of COUP-TFII antibody (Active Motif, CA 61213). Libraries were sequenced by Active Motif as single-end 75-mers on an Illumina NextSeq 500 instrument.

## ChIP-seq data analysis
Raw reads were preprocessed with TrimGalore v0.6.7[81] (parameters: --quality 20 --illumina --stringency 5 --length 30) for quality and adapter trimming. Reads were then mapped to the mm10 reference genome via Bowtie v2.1.0[82] using the "--local" setting. Alignments were filtered for a minimum MAPQ score of 5, then duplicates were removed by MarkDuplicates.jar (Picard tool suite v1.110[83]) with the REMOVE_DUPLICATES = TRUE option. Global similarity of two biological replicates was confirmed by calculating the Pearson R for the depth-normalized signal in 2,730,835 non-overlapping 1 kb bins tiled across the mm10 reference genome. The two replicate samples were merged prior to calling peaks, which was performed by the HOMER v4.11.1[84] 'findPeaks' function with parameters "-style factor -fdr 1e-5". Peak regions were subsequently re-sized to a width of 300 bp centered on the called peak midpoints. Peaks overlapping with mm10 blacklist regions were excluded from downstream analysis. Heatmap and metaplot views of the ChIP-seq data were generated based on counts of mapped read midpoints (after extension to estimated fragment length of 200 bp) in 20-bp bins tiled across regions ±2 kb of each peak, with quantifications normalized to 10 million uniquely-mapped fragments per condition (IP

or input). The genomic context of the ChIP-seq peaks was assigned according to distance to the nearest annotated TSS among NCBI RefSeq Curated gene models as downloaded from the UCSC Table Browser on April 21 2021. Enriched motif analysis was performed by the HOMER v4.11.1 'findMotifsGenome.pl' function using parameter "-size given".

## RNA extraction for bulk RNA-seq

Three E14.5 testes (control or *Nr2f2* cKO) were pooled per sample, for a total of 6 samples per group. RNA was extracted using the Arcturus™ PicoPure™ RNA Isolation Kit (Applied Biosystems, KIT0204). Briefly samples were dissociated in 100 µl of in extraction buffer with a microtube homogenizer and incubated at 42 °C for 30 min, then frozen at −80 °C for at least one hour. RNA was precipitated with 100 µl of 70% ethanol and loaded into a pre-conditioned column. Samples were centrifuged and the column was washed with Wash Buffer 1, followed by a step of DNA removal using the RNase-Free DNase Set (Qiagen, #79254). Samples were washed twice with Wash Buffer 2 and the columns were centrifuged to eliminate residual buffer. Samples were eluted in 12 µl of elution buffer. RNA was quantified using Qubit™ RNA high sensitivity assay kit (ThermoFisher Scientific Q32852). RNA quality was assessed using RNA ScreenTape (Agilent Technologies, #5067-5576) on a TapeStation 4200 (Agilent Technologies).

## Bulk RNA-seq

Two hundred and fifty ng of RNA were diluted to 50 µL in nuclease-free water and RNA libraries were prepared according to the TruSeq Stranded mRNA Library Prep protocol (Illumina, #20020594). High-Prep PCR beads were used for cleanup steps (MagBio Genomics, #AC-60050). SuperScript II Reverse Transcriptase was used for second strand synthesis (Invitrogen #100004925). cDNA was amplified with 14 PCR cycles. Library quality was checked using D1000 ScreenTape (Agilent Technologies, #5067-5582) on a TapeStation 4200 (Agilent Technologies). Library concentration was measured using a Qubit dsNA HS Assay Kit (Invitrogen/Thermo Fisher Scientific, Q32854) on a QubitFlex Fluorometer (Invitrogen/Thermo Fisher Scientific). Libraries were sequenced by the NIEHS Epigenomics and DNA Sequencing Core as paired-end 151-mers on an Illumina NextSeq 500 instrument.

## Bulk RNA-seq analysis

Raw read pairs were preprocessed with TrimGalore v0.6.7[81] (parameters: --quality 20 --illumina --stringency 5 --length 20 --paired) for quality and adapter trimming. Reads were then mapped to the mm10 reference genome via STAR v2.5[85] with parameters "--outSAMattrIHstart 0 --outFilterType BySJout --alignSJoverhangMin 8 --limitBAMsortRAM 55000000000 --outSAMstrandField intronMotif --outFilterIntronMotifs RemoveNoncanonical". Counts per gene were determined via featureCounts (Subread v1.5.0-p1)[86] with parameters "-s2 -Sfr -p". Evaluated gene models were taken from the NCBI RefSeq Curated annotations as downloaded from the UCSC Table Browser on April 21 2021. DESeq2 v1.28.1[87] was used for principal component analysis and identification of differentially expressed genes (threshold at FDR 0.05). Gene ontology, and pathway analysis, including reactome and wikipathways were analyzed using Enrichr[88–90]. Venn diagrams were generated using deepvenn[91].

## Bulk RNA-seq empirical projection

Heatmaps illustrating relative expression patterns in the snRNA-seq assay at genes identified as KO-vs-WT DEGs in the bulk RNA-seq dataset were generated with ComplexHeatmap v2.18.0[92]. For up- or down-regulated genes, the per-cluster average expression scores (as reported by Seurat's AverageExpression function) were scaled and then row-clustered via hclust. As the smallest cluster (c13) contains only two cells, it was ignored in this analysis. Gene groups were established by setting k in the color_branches function (dendextend v1.17.1[93]) to the largest k such that no resulting group included less than 2% of the query genes. In the heatmap views, the resulting gene groups and row-ordering were applied to the per-cell scaled expression scores reported by Seurat's ScaleData function (with regressing out of cell-cycle differences and % mitochondrial content). For clarity in visualization, the width of the smaller clusters (c8 through c12) are expanded.

## Reporting summary

Further information on research design is available in the Nature Portfolio Reporting Summary linked to this article.

## Data availability

Single-nucleus multiomic data have been deposited at GEO: GSE275901. NR2F2 ChIP-seq data have been deposited at GEO: GSE275900. Nr2f2 cKO Bulk RNA-seq data have been deposited at GEO: GSE275902. Other requests for data may be directed to and will be fulfilled by the Lead Contact, Humphrey Yao. Source data are provided with this paper.

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

## Acknowledgements

We are grateful to the NIEHS Molecular Genomics Core, in particular Xin Xu, Erin Smithberger, Jason Malphurs and Gregory Solomon for the RNA-seq processing and Comparative Medicine Branch for mouse colony maintenance. We would like to thank Yu-Ying Chen and Adriana Alexander for the generation of the multiomic shiny app. This work was supported by the Intramural Research Program (ES102965 to H.H.-C.Y.) of the NIH, National Institute of Environmental Health Sciences.

## Author contributions

M.A.E. and H.Y. conceived the project and designed the experiments. M.A.E., A.S.K., K.F.R., P.R.B., K.M. and C.M.A. conducted experiments, whereas S.A.G. carried out the bioinformatics. M.A.E. and H.Y. wrote the manuscript. All authors edited and revised the text.

## Funding

## Competing interests

The authors declare no competing interests.
