## [Transparent Peer Review file · Nature Communications]

NR2F2 Regulation of Interstitial Cell Fate in the Embryonic Mouse Testis and Its Impact on Differences of Sex Development

Corresponding Author: Dr Humphrey Yao

Version 0:

Reviewer comments:

Reviewer #1

(Remarks to the Author)

In this manuscript, Yao and colleagues utilized single cell multimoics analysis and mouse lineages tracing tools to study fetal Leydig cells, revealing the critical role of NR2F2 in regulating fetal Leydig cell development. Overall, the manuscript is well written with clear logic, and provided a good collection of evidence to support the main conclusions. Here, this reviewer feels that the following concerns should be considered before the manuscript can be proceeded to formal publication.

1. The article does not quantify the number of cells in the staining images, including Figures 2B-D, 3, 4C, and 5B.
2. In the single-nucleus RNA-seq/ATAC-seq data, the authors identified clusters c0, c1, c5, c7, and c10 as "potential interstitial cells" (Line 296). Are *Pdgfra* or other interstitial marker genes expressed in these clusters?
3. Cells in cluster c5 express *Podxl* (Line 298), a specific marker gene for podocytes in the mesonephros. To determine if c5 cells are testicular cells, *PODXL* staining is necessary. If they do not express this marker, they should be excluded from further analysis.
4. The authors state that "fetal Leydig cells (c3) clustered closely to the c0 progenitors in the ATAC-seq UMAP" (Line 318). However, the distance between clusters on a UMAP does not represent their transcriptomic similarity. To assess the similarity between c0 and c3, a more appropriate method should be employed.
5. The authors refer to c0 as "progenitors" (Line 318). What is the basis for this conclusion? Are genes associated with interstitial progenitors, such as cell cycle genes, *Tcf21*, or *Wnt5a*, expressed in these cells? If the authors intend to explore the developmental trajectory between c0 and other interstitial cells, they should consider using more precise analytical methods, such as pseudotime analysis.
6. The reference to "Fig. S3B" in Line 405 should be corrected to "Fig. S4B."
7. The statement that "loss of *Nr2f2* affects interstitial, fetal Leydig cells, immune cells, and germ cells" (Line 454) lacks supporting evidence. The authors should present relevant staining results, such as combined *KI67/PDGFR α* staining in both *Nr2f2* knockout and control testes, along with quantification of *KI67⁺/PDGFR α ⁺* cells.
8. The authors assert that NR2F2 regulates steroidogenic genes in fetal Leydig cells (Lines 542-544). However, since NR2F2 is not expressed in fetal Leydig cells, this conclusion is unconvincing. I recommend that the authors include positive and negative controls for the ChIP-seq experiments.

Reviewer #2

(Remarks to the Author)

In this study, the authors used single nucleus multiome analysis to reveal the heterogeneity of fetal testicular interstitial cells.

They identified significant differences in gene expression between fetal Leydig cells and other interstitial cell populations. Through ATAC-seq analysis, they identified cluster C0 as a progenitor of fetal Leydig cells. Focusing on Nr2f2 expression in C0, the authors performed immunohistochemical analysis of Nr2f2 expression, lineage tracing of Nr2f2-expressing cells, and conditional Nr2f2 knockout analyses. They concluded that Nr2f2 plays an important role in fetal Leydig cell differentiation. Furthermore, ChIP-seq analysis of NR2F2 identified its target genes, although much remains unknown about the downstream gene regulatory network.

The reviewer has the following concerns:

1. While the authors identified cluster C0 as a progenitor population of fetal Leydig cells, Nr2f2 is also expressed in other cell populations, such as those in the tunica albuginea, myoid cells, pericytes, and mesonephros, and the role of NR2F2 in these populations is unclear. This raises the possibility of heterogeneity within NR2F2-positive cells, suggesting that the phenotype observed in Nr2f2 knockout mice may not solely reflect the role of Nr2f2 in Leydig progenitors. Although this point is discussed, no specific analyses are provided to support it. Additionally, trajectory analysis would be useful as supporting data to demonstrate that C0 is a progenitor population of fetal Leydig cells.
2. Regarding the single-cell analysis data, the authors do not show the expression of known interstitial cell markers, such as Nr2f2, Arx, Pdgfra, Tcf21, and Wnt5a. Demonstrating Nr2f2 expression across clusters is essential for interpreting the knockout phenotype.
3. The authors used Wt1-CreERT2 to conditionally knock out Nr2f2 in the testes. However, WT1 is primarily expressed in undifferentiated Sertoli cells, making it a suboptimal choice for deleting Nr2f2 in interstitial cells. Indeed, while Sun1-GFP expression was detected in many Sertoli cells, only a subset of interstitial NR2F2-expressing cells showed GFP expression. Quantitative data on knockout efficiency are needed.
4. Despite the presence of many HSD3B1-expressing fetal Leydig cells in Nr2f2 cKO mice, significant external genital abnormalities were observed, creating a paradox. Gene expression analysis in the testes and quantification of androgen levels in cKO mice are necessary. Despite extensive single-cell analysis at the start of the paper, there is no single-cell analysis of the cKO phenotype, making the conclusions less well supported. Additionally, analysis of cellular morphological changes at both the light and electron microscopy levels would be useful.
5. A major unresolved issue is that while Nr2f2 is hypothesized to suppress fetal Leydig cell differentiation, its knockout leads to impaired differentiation. Although the authors propose various hypotheses based on ChIP-seq and transcriptome data, the precise mechanism remains elusive, leaving the reader with an incomplete understanding.
6. It has been reported that Tmsb10 triggers the differentiation of fetal Leydig from progenitor cells in the fetal testicular interstitium. The authors should mention whether the C0 include Tmsb10-positive cells or not.
7. On a minor note, in the schematic figure (Fig. 7), fetal Leydig cells are depicted with cellular polarity, which seems inappropriate. Fetal Leydig cells are interstitial and do not typically exhibit unidirectional cellular projections.

In conclusion, while this manuscript provides valuable insights into the mechanisms of fetal Leydig cell differentiation, many important questions remain unsolved.

Reviewer #3

(Remarks to the Author)

Leydig cells are critical for the development of primary and secondary sex characteristics, primarily through the synthesis of steroid hormones. Despite their importance, the origin and characterization of steroidogenic precursors in the developing testis and the factors that regulate their differentiation into fetal and adult Leydig cells are not well understood. This manuscript aims to address these areas, in particular the role of NR2F2 in the transition of testicular interstitial progenitor cells into fetal Leydig cells. The authors integrated single-cell multiomics and bulk RNA sequencing analyses with *in vivo* mouse lineage tracing and functional genomics to first characterize the interstitial cell populations present in the mouse fetal testis. Their findings also highlight the essential role of the NR2F2 transcription factor in regulating the initiation and progression of fetal Leydig cell differentiation within steroidogenic precursors. However, given the transcriptomic plasticity of steroidogenic progenitors and their rapid differentiation into FLCs, it is crucial to extend the multiomic analyses to stages earlier than E14.5 and to ensure that the embryonic stages selected for analysis are consistent across the different experimental approaches used. In addition, the transcriptomic, lineage tracing and loss-of-function analyses appear in some cases to be oversimplified and lack adequate technical validation, which undermines the reliability of the results and conclusions presented in the manuscript.

Main comments:

- Regarding the terminology for Leydig progenitor cells, the term "non-steroidogenic interstitial cells" is too broad, as various somatic cells could fall under this category. Additionally, although these progenitors are not themselves steroidogenic, they belong to a lineage where the differentiated Leydig cells primarily perform steroidogenesis. A more accurate and descriptive term would be "steroidogenic progenitors" or "Leydig cell progenitors", as this better reflects their role in the cell lineage.
- It would be beneficial to mention, either in the introduction or discussion, the results of a recent study on murine steroidogenic lineages, that combined single-cell RNA sequencing (scRNA-seq) and lineage tracing (Ademi et al., 2022). This study showed that steroidogenic progenitors differentiate into fetal Leydig cells (FLCs) within a narrow developmental window, specifically around embryonic days E12.5 and E13.5. By E16.5, these progenitors predominantly give rise to peritubular myoid cells, while the remaining population later differentiates into adult Leydig cells (ALCs). In addition, the study identified a transient population of pre-Leydig cells at E12.5 and E13.5, which were characterized transcriptionally.
- The first section of the manuscript, which describes the characterization of interstitial cell populations in the embryonic testis, is crucial but would benefit from further analysis. Given the significant plasticity of steroidogenic progenitors and the presence of intermediate cell types at E12.5 and E13.5 (e.g. pre-LCs), it is recommended that the multiomic single-cell

analysis be extended to earlier developmental stages, particularly E11.5, E12.5, and E13.5, to provide a more comprehensive view of the lineage differentiation processes. The E14.5 stage is too late to assess the early stages of steroidogenic progenitor differentiation into FLCs. This dual approach would provide insight into the regulatory mechanisms driving the transition from progenitor cells to fetal Leydig cells (FLCs) and the emergence of other intermediate cell types.

- On lines 296-308, the authors characterized the interstitial compartment of the mouse testis using single-nucleus multiomics data, and identified five distinct clusters. They confirmed the annotation and localization of different cell populations through immunofluorescence, specifically detecting ALDH1A2+ testicular epithelial cells, ITGA8+ mesonephric cells, and LGI1+ pericytes. These findings are significant and should be included in one of the primary figures in the manuscript. It is important to note that MYLK1 is expressed in several groups of interstitial cells, so it should not be considered as a specific marker for group 1. Furthermore, the expression profiles of *Acta2* and *Myh1* suggest that cluster 1 consists predominantly of peritubular myoid cells.
- As noted in lines 326-329, *Nr2f2* is highlighted in the differential expression analysis and used to lineage trace steroidogenic progenitors; however, there is no UMAP or any figure illustrating the expression of *Nr2f2* in the single nuclei data. Including such a visual representation would enhance the clarity and impact of the findings regarding the role of *Nr2f2* in these progenitors.
- The authors used *Nr2f2-CreER;CAGSun1/sfGFP* reporter mice to track the fate of NR2F2+ cells in the mouse testis in an inducible manner. Samples were collected at E14.5 following three different tamoxifen (TAM) injection protocols: 1) one injection at E11.5, 2) one injection at E12.5, and 3) one injection at E13.5. While the strategy to follow the fate of NR2F2+ cells at E14.5 is relevant, we strongly recommend that the authors evaluate recombination efficiency 24 hours after each injection, as they did not report the proportion of NR2F2+ cells targeted by their transgenic system. The low proportion of GFP+ cells within the NR2F2+ population 24 hours after TAM injection at E13.5 highlights a major limitation of the system, which appears unsuitable for labeling a substantial proportion of NR2F2+ cells after a single injection.
- The authors claim that cluster c0 represents progenitors of cluster c3 at E14.5; however, their lineage tracing was performed with inductions at E11.5, E12.5, and E13.5, all preceding E14.5. This temporal discrepancy complicates the conclusion that the observed differences in transcriptomes and chromatin accessibility between c0 and c3 accurately reflect the processes involved in fetal Leydig cell differentiation. To improve the characterization of the progenitors associated with the lineage tracing experiments, it would be beneficial to generate multiomic data at the appropriate developmental time points.
- It is not possible to conclude that differentiation and the production of fetal Leydig cells (FLCs) is a continuous process during testis development, as suggested in lines 361, 366, and 643. Induction at E11.5, E12.5, and E13.5 alone is not sufficient to support this conclusion. Lineage tracing studies (Ademi et al 2022) have shown that the vast majority of both FLCs and ALCs are derived from steroidogenic progenitors that are primarily specified around E11.5. Only a small fraction of steroidogenic progenitors is specified later, and they contribute minimally to the overall FLC and ALC populations.
- We suggest that the authors invert Figures 2 and 3 to improve the readability of the manuscript. Presenting the NR2F2 localization in the developing mouse gonad first would provide a clearer context before moving on to the lineage tracing experiments that follow the fate of NR2F2+ putative steroidogenic progenitors.
- Concerns regarding the use of the *Wt1:CreERT* transgenic line: The tamoxifen-inducible Cre system is commonly used to control Cre-mediated recombination at specific times and in specific cell types. However, tamoxifen is not a neutral agent in this process; it is a well-documented endocrine disruptor known to cause adverse effects on the testis and the reproductive endocrine system (see doi: 10.1038/s41598-017-09016-4). In addition, the *Wt1:CreERT* transgene is a knock-in, which could lead to *Wt1* haploinsufficiency. When combined with tamoxifen exposure, this could lead to alterations in testicular development. It is important to know whether the authors included adequate controls to account for these factors and what they were?
- We also recommend that the authors use the lineage tracing strategy in the NR2F2 cKO model to detect ARX/LacZ and confirm the NR2F2-independent expression of ARX. In addition, the detection of LacZ/HSD3B1 could help to determine whether NR2F2 is essential for Leydig cell differentiation. This would strengthen the conclusions regarding the role of NR2F2 in the fate of steroidogenic progenitors.
- Interstitial progenitors have previously been shown to be associated with both the steroidogenic lineage and the peritubular myoid cell (PMC) lineage. However, the authors conclude that "a severely affected population in the *Nr2f2* cKO testes was the smooth muscle cells, particularly those in the tunica albuginea" (Lines 676-677). Given that the study focuses primarily on testis development up to E14.5, a stage before the full differentiation of PMCs, it would be insightful for the authors to evaluate the contribution of NR2F2+ cells to PMC differentiation. This could be achieved by co-labeling GFP with SMA at E17.5, when PMCs are more fully differentiated, to assess the role of NR2F2+ cells in PMC development.
- The term "bulk deconvolution" is incorrectly used in the manuscript. What the authors describe is not true deconvolution but rather an analysis of differentially expressed genes (DEGs) between cKO and WT which are then visualized across different single nucleus (SN) clusters. This is better described as an "empirical projection" rather than deconvolution. True deconvolution involves constructing reference signature matrices from single-cell data and projecting the bulk data onto these matrices using various statistical frameworks to estimate cell type proportions (see <https://doi.org/10.1038/s41467-020-19015-1> for more details). A more rigorous deconvolution approach could provide more robust insights into potential shifts in cell population proportions. This is particularly relevant in the context of bulk RNA-seq, as it allows for a more accurate representation of changes in cellular composition.
- Regarding the bulk RNA seq analysis (line 465), I have concerns about how the term "most affected population" is defined. This assessment cannot be reliably made visually on a heatmap, where color variations are relative to gene expression changes across the nuclei. Such an analysis only indicates that certain genes expressed in specific subpopulations are differentially expressed in the bulk, which lacks rigor. A more informative approach would be to quantify the contribution of each gene to each cluster and then to calculate the effect of the observed expression variations in differentially expressed genes (DEGs). The authors should exercise caution in making such claims.
- While gene groups can to some extent be associated with specific populations, their up or down-regulation in the cKO can also be linked to a reduction in population size rather than an alteration in its transcriptome. Standard bulk deconvolution

utilizing analytical tools like MUSiC could yield valuable insights but would eventually fail at clarifying what is really changing in the cKO: population proportions or their transcriptome. Manual quantification of the possibly affected populations or performing single-nucleus multiome analyses on cKO testes would provide a clearer understanding of the underlying changes in the mutant model.

- Single-cell and ChIP-seq data should be accompanied by the QC plots.
- ChIP-seq should be accompanied by metaplots and heatmaps of the peak coverage profile/distribution to see the quality and pattern of the signal to objectively evaluate the quality of the data (for ex., DOI: 10.26508/lsa.202201854, Fig 1).

Minor Comments:

- Line 79: for Wnt5a+ cells, this also includes adult Leydig cells
- Line 156: Transgene names should be italicized.
- Line 166: Start the sentence with "Five" instead of using the number "5".
- Line 186: Clarify what "0.3% acid alcohol" refers to (e.g., specify the composition of the solution).
- Lines 296-308: The authors characterized the interstitial compartment of the mouse testis using single-nucleus multiomics data, and identified five distinct clusters. They confirmed the annotation and localization of different cell populations through immunofluorescence, notably detecting ALDH1A2+ testicular epithelial cells, ITGA8+ mesonephric cells, and LGI1+ pericytes. These findings are significant and should be integrated into one of the primary figures in the manuscript. MYLK1 is expressed across several groups of interstitial cells, so it should not be regarded as a specific marker for Group 1. Furthermore, the expression profiles of Acta2 and Myh1 suggest that cluster 1 predominantly consists of peritubular myoid cells.
- Line 405: Replace Fig S3B by Fig S4B
- Line 419: Replace muscle cells with contractile cells
- Line 515: Replace "majority" with "the biggest part".
- Regarding Tables S6, to improve readability and usability, I recommend adding descriptive titles directly within the document. Additionally, please ensure that the filenames of the tables are consistent and clearly labeled. Furthermore, some column names in Table S6 lack clarity, making it challenging to infer the data presented.
- Lines 656-657: The statement is inaccurate because the characterization of the different cell populations in the testis using a gonadal atlas with scRNA sequencing was already published two years ago. This study described two populations of endothelial cells and identified a population of immune cells.

Reviewer #4

(Remarks to the Author)

Version 1:

Reviewer comments:

Reviewer #1

(Remarks to the Author)

The authors have taken into consideration the suggestions raised in the previous review process and have made appropriate revisions. The changes made have improved the quality of the manuscript. I have no further questions with the revised manuscript.

Reviewer #2

(Remarks to the Author)

To the Authors

Thank you for your sincere response to the reviewers' comments. I feel that many of the points raised in the initial review have been addressed satisfactorily, either through new data or reanalysis of existing data. Regarding the revised manuscript, the reviewer would like to highlight the following two points:

1. In Supplemental Figure 2C, the results of the trajectory analysis are presented. According to this figure, C0 is suggested to be the progenitor of C3 (fetal Leydig cells), but a differentiation pathway from C1 to C0 is also observed. Since C1 expresses Mylk, Myh11, and Acta2, it can be identified as a contractile cell. Considering this, along with the disappearance of the tunica albuginea observed in Nr2f2-cKO mice, could it be interpreted that the reduction in the supply of fetal Leydig progenitors from the tunica albuginea leads to a decrease in both interstitial cells and fetal Leydig cells? In fact, many of the genes whose expression is reduced in Nr2f2-cKO mice appear to be associated with ECM and contractile function. If the above-mentioned mechanism can also be considered, even partially, alongside the gene expression-mediated mechanism proposed by the authors, how about incorporating it into the discussion?

2. For testosterone measurements in fetal testes, especially in the small testes of cKO mice, I believe that mass spectrometry is the most appropriate method. While I am unable to provide specific names, there are researchers and companies that measure steroid hormones using mass spectrometry on blood or tissue samples. To explain the dramatic phenotype observed in the external genitalia, showing only the number of Leydig cells seems insufficient. To demonstrate the functional capacity of the remaining Leydig cells compared to normal Leydig cells, testosterone measurement would be ideal. If this is

challenging, histological analyses could serve as supplementary data. Observing lipid droplets and rough endoplasmic reticulum using electron microscopy would be ideal, but if this is also difficult, perhaps optical microscopy-level observations or Oil Red O staining could be attempted.

Reviewer #3

(Remarks to the Author)

Most of my questions and concerns have been addressed. However, two points remain unresolved, and merit further consideration.

Before addressing the two remaining issues, I would like to underscore some quantitative and temporal information regarding steroidogenic progenitors and Leydig cells that are currently missing from the manuscript's introduction but are crucial for providing context. This information should be incorporated into the introduction in some form. In mice, the first Leydig cells appear toward the end of E12.5, with their numbers increasing substantially from E13.5 onwards. This timeline suggests that the Nr2f2+/Wnt5a+/Arx+ steroidogenic progenitors at the origin of fetal Leydig cells must be specified during an earlier developmental window. This observation is supported by studies combining scRNA-seq and quantitative lineage tracing, which demonstrated that Wnt5a+/Nr2f2+/Arx+ steroidogenic progenitors specified between E11.5 and E12.5 contribute to the majority of fetal Leydig cells (74%) and adult Leydig cells (79%) (Ademi et al 2022). These findings indicate that the majority of both fetal and adult Leydig cells originate from Wnt5a+/Nr2f2+/Arx+ steroidogenic progenitors primarily specified around E11.5–E12.5. In contrast, only a minor fraction of steroidogenic progenitors are specified later, and these contribute minimally to the overall populations of fetal and adult Leydig cells.

Comment 3: The first two chapters of the results section are critical in establishing context and characterizing the interstitial populations of the testis, particularly the Nr2f2+ steroidogenic progenitors. A significant concern that remains inadequately addressed pertains to the specification of these steroidogenic progenitors and their considerable plasticity during the critical developmental window between E11.5 and E13.5. By focusing solely on the E14.5 stage, the manuscript misses key insights into how NR2F2+ steroidogenic progenitors are initially specified, their transcriptional evolution, and their subsequent differentiation into fetal Leydig cells. Expanding the analysis to include earlier stages, particularly E12.5 and E13.5, would greatly enhance the understanding of these key developmental processes.

Given that multi-omic data for these earlier stages are available, I strongly recommend incorporating scRNA seq data from E12.5 and E13.5 to comprehensively analyze the emergence and evolution of steroidogenic progenitors between E12.5 and E14.5. Including these data would substantially enhance the depth and overall impact of the study, especially for a manuscript intended for publication in Nature Communications.

Comments 2, 8: There seems to be a tendency to underrepresent or overlook prior contributions from other laboratories. Ensuring accurate and comprehensive citation of existing work is crucial, as it does not diminish the value of the authors' contributions in this study but rather contextualizes their findings within the broader field.

While the authors incorporated findings from Ademi et al. (2022) into the revised manuscript, as suggested in the initial review (Lines 525–533), the current phrasing is inaccurate and could be improved. The conclusions of Ademi et al. are supported by multiple lineage-tracing experiments induced at various developmental stages.

Additionally, there seems to be a misunderstanding in the rebuttal letter. The authors state: "If fetal Leydig cells were specified at E11.5, as suggested by Ademi et al.,....". However, Ademi et al. did not claim that FLCs themselves are specified at E11.5. Rather, the study clarified that Wnt5a+/Nr2f2+/Arx+ steroidogenic progenitors—precursors to fetal Leydig cells—are specified during this period, and these progenitors subsequently differentiate into fetal Leydig cells from E12.5 onward. It would be beneficial for the readers to explicitly state in the introduction that the majority of both fetal and adult Leydig cells are derived from steroidogenic progenitors primarily specified around E11.5–E12.5.

Finally, I recommend adding a discussion of a recently published study

(<https://www.biorxiv.org/content/10.1101/2024.07.17.602099v1>), which presents similar findings. Acknowledging this work would further contextualize the study's contributions and strengthen its impact.

Reviewer #4

(Remarks to the Author)

Version 2:

Reviewer comments:

Reviewer #2

(Remarks to the Author)

I appreciate the authors' sincere responses to my comments. I consider the manuscript worthy of acceptance.

Reviewer #3

(Remarks to the Author)

The authors have adequately addressed the remaining pending comments.

Response to the reviewers:

We thank the reviewers for their helpful comments, which allow us to improve the quality of the manuscript. The manuscript lines referred in the answers are based on the “clean” version of the manuscript, without the edits.

Reviewer 1:

1. The article does not quantify the number of cells in the staining images, including Figures 2B-D, 3, 4C, and 5B.

Response: As the purpose of Fig. 3 (now Fig. 2) is to evaluate qualitatively the expression pattern and the colocalization of NR2F2 with different cell population markers, quantification of cell numbers does not add information pertinent to the question. Similarly, in Fig. 2 (now Fig. 3), the purpose of the experiment was to investigate the fate of the NR2F2 positive cells using lineage tracing at different developmental timepoints. Due to the known variability of tamoxifen-induced lineage tracing models, it is difficult to discern whether differences in the numbers of labeled cells were the result of restricted developmental trajectories or variations in tamoxifen action. For this reason, we only reported the positive or negative labelling cells of each cell type. In figures 4C, we now provide the quantification of the interstitial and gonadal area (Supplementary Fig. 4B-G), the number of interstitial cells in wildtype (Nr2f2 positive) or knockout (LacZ positive) testis, and the number of proliferative interstitial cells positive for LacZ or Nr2f2 (Supplementary Fig. 6C & D). Additionally, we now provide quantification of fetal Leydig cells in both control and knockout gonads at E14.5 (Supplementary Fig. 4G) and E17.5 (Fig. 5C & D).

2. In the single-nucleus RNA-seq/ATAC-seq data, the authors identified clusters c0, c1, c5, c7, and c10 as "potential interstitial cells" (Line 296). Are *Pdgfra* or other interstitial marker genes expressed in these clusters?

Response: We apologize for not being clear in this section. All the “interstitial clusters” were positive for *Pdgfra*, *Wnt5a*, *Tcf21* and *Arx*, proposed markers of the interstitial cell populations. This is now reflected in Supplementary Fig. 2A and in the text (lines 113-115)

3. Cells in cluster c5 express *Podxl* (Line 298), a specific marker gene for podocytes in the mesonephros. To determine if c5 cells are testicular cells, *PODXL* staining is necessary. If they do not express this marker, they should be excluded from further analysis.

Response: Recent findings indicate that *PODXL* is expressed not only in podocytes within the mesonephros but also in the gonadal epithelium in humans (Taelman, Jasin et al., *Developmental Cell*, 2024, 10.1016/j.devcel.2024.01.006). This aligns with our single-cell RNA-seq data, where *Podxl* is expressed in the epithelial cluster (c1), which also

shows *Aldh1a2* expression (Fig. 1D). We have included a brief discussion of this in the manuscript text (Lines 125-126). Unfortunately, neither us nor our collaborators have an antibody for PODXL.

4. The authors state that "fetal Leydig cells (c3) clustered closely to the c0 progenitors in the ATAC-seq UMAP" (Line 318). However, the distance between clusters on a UMAP does not represent their transcriptomic similarity. To assess the similarity between c0 and c3, a more appropriate method should be employed.

Response: To assess the chromatin similarity between c3 and c0 we identified the 2000 ATAC peaks with the most variable signal across all cells, averaged the signal for cells per assigned cluster, calculated the Euclidean distance between each cluster and generated a dendrogram of clusters from the distance matrix (Supplementary Fig. 2B). Since we cannot infer directly from the dendrogram which cluster is closest to c3 (it could be c0 / c1 / c5 or c7), we have reported the Euclidean distance calculated between c3 and each of the other clusters, which does indicate that c0 is the cluster most similar to c3 (Supplementary Fig. 2B). We included this results in the text (Lines 140-146), as well as in the methods section (Lines 702-706)

5. The authors refer to c0 as "progenitors" (Line 318). What is the basis for this conclusion? Are genes associated with interstitial progenitors, such as cell cycle genes, *Tcf21*, or *Wnt5a*, expressed in these cells? If the authors intend to explore the developmental trajectory between c0 and other interstitial cells, they should consider using more precise analytical methods, such as pseudotime analysis.

Response: As all the "interstitial" cells (c0, c1, c5, c7, c10) express the conventional interstitial progenitor markers (*Tcf21*, *Wnt5a*, *Arx*, *Pdgfra*), we identified the c0 as the fetal Leydig cell progenitor cells, based on the chromatin accessibility similarity in our single nuclei ATAC-seq data. We validated the similarity by showing that c0 has the shortest Euclidean distance to c3 (See comment 4 response, Supplementary Fig. 2B). In addition, we performed trajectory analysis, as suggested, using Monocle3 on the single nuclei ATAC-seq data, showing that c0 and c3 are connected by a trajectory path (Supplementary Fig. 2C). This suggests that c0 is developmentally related to c3. We included this results in the text (Lines 148-151), as well as in the methods section (Lines 701-702)

6. The reference to "Fig. S3B" in Line 405 should be corrected to "Fig. S4B."

Response: This is corrected in the new version of the manuscript.

7. The statement that "loss of *Nr2f2* affects interstitial, fetal Leydig cells, immune cells, and germ cells" (Line 454) lacks supporting evidence. The authors should present relevant staining results, such as combined KI67/PDGFR α staining in both *Nr2f2* knockout and control testes, along with quantification of KI67+/PDGFR α + cells.

Response: Since the RNA-seq “empirical projection” indicated that *Nr2f2* knockout interstitial cells downregulated genes associated with proliferation, particularly those involved in the G2/M transition (Supplementary Fig. 5A), we chose to assess cell proliferation using pH3. This marker specifically labels cells in late G2 and M phases, providing greater specificity than Ki-67, which also marks cells in G1, S, and G2 phases. Because LacZ expression is activated upon *Nr2f2* deletion, we were able to distinguish and quantify proliferation in both control (NR2F2-positive) and *Nr2f2* knockout (LacZ-positive, NR2F2-negative) interstitial cells. This analysis revealed a general reduction in interstitial cell proliferation in knockout gonads compared to controls (Supplementary Fig. 6B-C). Furthermore, within the knockout gonads, cells lacking *Nr2f2* exhibited lower proliferation rates than those retaining *Nr2f2* (Supplementary Fig. 6D). These findings suggest that *Nr2f2* is essential for interstitial cell proliferation, which likely contributes to the observed reductions in testicular size (Supplementary Fig. 4B), interstitial area (Supplementary Fig. 4C), and interstitial cell numbers (Supplementary Fig. 4E). We have included these results in the text (lines 321-330).

8. The authors assert that NR2F2 regulates steroidogenic genes in fetal Leydig cells (Lines 542-544). However, since NR2F2 is not expressed in fetal Leydig cells, this conclusion is unconvincing. I recommend that the authors include positive and negative controls for the ChIP-seq experiments.

Response: We apologize for the lack of clarity. *Nr2f2* is indeed not expressed in fetal Leydig cells. What we intended to convey is that, in interstitial cells, NR2F2 inhibits steroidogenic genes based on our ChIP-seq data that showed NR2F2 binding to regulatory regions of genes associated with fetal Leydig cells, such as *Nr5a1* and *Star*. These genes are not expressed in interstitial cells (where NR2F2 is present) but expressed in fetal Leydig cells (where NR2F2 is absent).

Based on this, we infer that NR2F2 represses the expression of a subset of Leydig cell-specific genes in interstitial cells. During Leydig cell differentiation, the loss of NR2F2 likely leads to the de-repression of these genes, allowing their activation, potentially through the binding of other transcriptional regulators to the same regions. We have revised the manuscript to clarify this explanation (Lines 424-433). Regarding our ChIP-seq data, our negative control was the input, confirming that the enrichment of the targeted regions was specific to the antibody (Supplementary Fig. 8A). While we did not use a positive control, both *de novo* and known motif analyses conducted with JASPAR identified NR2F2 as the top candidate, providing strong evidence that we successfully captured NR2F2 targets (Fig. 6B and C). We have now included the quality control (QC) metrics for our ChIP-seq data in the manuscript (Supplementary Fig. 8).

Reviewer #2 (Remarks to the Author):

1. While the authors identified cluster C0 as a progenitor population of fetal Leydig cells, *Nr2f2* is also expressed in other cell populations, such as those in the tunica albuginea, myoid cells, pericytes, and mesonephros, and the role of NR2F2 in these populations is unclear. This raises the possibility of heterogeneity within NR2F2-positive cells, suggesting that the phenotype observed in *Nr2f2* knockout mice may not solely reflect the role of *Nr2f2* in Leydig progenitors. Although this point is discussed, no specific analyses are provided to support it. Additionally, trajectory analysis would be useful as supporting data to demonstrate that C0 is a progenitor population of fetal Leydig cells.

Response: We agree with the reviewer that *Nr2f2* is expressed in different cell populations. This expression reflects the multiple phenotypes in the knockout that affects these populations. In particular, we addressed two independent phenotypes, one involving the lack of tunica albuginea (Fig. 4C and Supplementary Fig. 7B), suggesting that *Nr2f2* is required for a proper development of the tunica. Secondly, we discovered a reduced number of Leydig cells in testicular interstitium (Fig. 4C and 5B-D and Supplementary Fig. 4D). We expanded the multifactorial effect of *Nr2f2* in a new section focusing on the contractile cells (lines 369-390) and we changed the title to reflect a broader role of *Nr2f2* in interstitial cell populations (not only in fetal Leydig cells).

In addition, we performed trajectory analysis, as suggested, using Monocle3 in the single nuclei ATAC-seq data, showing that c0 and c3 are connected by a trajectory path (Supplementary Fig. 2C). This suggests that c0 is developmentally related to c3. We now include this in the text (Lines 148-151)

2. Regarding the single-cell analysis data, the authors do not show the expression of known interstitial cell markers, such as *Nr2f2*, *Arx*, *Pdgfra*, *Tcf21*, and *Wnt5a*. Demonstrating *Nr2f2* expression across clusters is essential for interpreting the knockout phenotype.

Response: We thank the author for this suggestion. All the “interstitial clusters” were positive for *Pdgfra*, *Wnt5a*, *Tcf21*, *Nr2f2* and *Arx*, known markers of the interstitial cell populations. This is now reflected in Supplementary Fig. 2A and in the text (lines 113-115).

3. The authors used *Wt1-CreERT2* to conditionally knock out *Nr2f2* in the testes. However, WT1 is primarily expressed in undifferentiated Sertoli cells, making it a suboptimal choice for deleting *Nr2f2* in interstitial cells. Indeed, while *Sun1-GFP* expression was detected in many Sertoli cells, only a subset of interstitial NR2F2-expressing cells showed GFP expression. Quantitative data on knockout deficiency are needed.

Response: Unfortunately, we did not have access to a more specific CreER mouse model to selectively knock out *Nr2f2* in interstitial cell progenitors. With the *Wt1-CreERT2* model, both interstitial and supporting cells are targeted (in Supplementary Fig. 3B).

However, *Nr2f2* is not expressed in Sertoli cells (NR5A1-positive) at the time points of tamoxifen injection (E11.5 and E12.5, Fig. 2). Furthermore, lineage tracing at these stages showed no labeling of Sertoli cells (Fig. 3). Therefore, we do not expect any significant impact of *Nr2f2* knockout in Sertoli cells. This is supported by our RNA-seq data, which shows no significant changes in the expression of key Sertoli cell markers such as *Sox9*, *Amh*, and *Dmrt1* in *Nr2f2* conditional knockouts (cKO). Using the *Wt1-CreERT2* model and two tamoxifen injections, we achieved an average *Nr2f2* knockout efficiency of 75.4% at E14.5, with a range of 67.3% to 86%, as determined by the proportion of NR2F2 and LacZ-positive cells. Cells positive for both NR2F2 and LacZ (single-allele knockouts) were counted as NR2F2-positive wild-type cells. We have included this data in the manuscript (lines 281-286, in Supplementary Fig. 4E).

4. Despite the presence of many HSD3B1-expressing fetal Leydig cells in *Nr2f2* cKO mice, significant external genital abnormalities were observed, creating a paradox. Gene expression analysis in the testes and quantification of androgen levels in cKO mice are necessary. Despite extensive single-cell analysis at the start of the paper, there is no single-cell analysis of the cKO phenotype, making the conclusions less well supported. Additionally, analysis of cellular morphological changes at both the light and electron microscopy levels would be useful.

Response: We believe that single-cell analysis of the *Nr2f2* cKO testes would provide information similar to our bulk RNA-seq and sequential “empirical projection.” Our bulk RNA-seq and sequential “empirical projection identified and validated the cell types that were affected in *Nr2f2* cKO samples and the potential pathways affected on each cell type. We agree with the reviewer regarding the highly penetrant external genital abnormalities, as all knockout samples exhibited defects. Immunostaining of genital cross-sections suggests that insufficient androgen action prevented urethral closure. Unfortunately, due to the small size of our samples, we were unable to detect testicular androgen levels using our methods available. We would be eager to quantify these androgen levels if the reviewer could recommend a kit suitable for small tissue samples, especially given the reduced size of the knockout testes. To address the issue of direct androgen quantification, we assessed Leydig cell numbers in both control and *Nr2f2* cKO testes. Our analysis revealed an average 50% reduction in Leydig cell density per testicular area (Fig. 5C). Additionally, due to the overall reduction in testis size in *Nr2f2* cKO samples (Supplementary Fig. 7A), the total number of Leydig cells per testis was approximately 70% lower than in controls (Fig. 5D), potentially reflecting a proportional decrease in steroidogenic capacity. We included this data in the manuscript (Lines 349-353). While we acknowledge that ultrastructural analysis of Leydig cells via electron microscopy could provide valuable insights, we consider this outside the scope of the current study.

6. It has been reported that *Tmsb10* triggers the differentiation of fetal Leydig from progenitor cells in the fetal testicular interstitium. The authors should mention whether the CO include *Tmsb10*-positive cells or not.

Response: We thank the reviewers for suggesting *Tmsb10*, which we had not previously considered in our analysis. *Tmsb10* is a differentially expressed gene (DEG) in cluster c0 but not in c3 (Supplementary Table 2) and is downregulated in *Nr2f2* cKO samples (Supplementary Table 3). Additionally, our ChIP-seq data revealed a significant *Nr2f2* binding peak at the transcription start site (TSS) of *Tmsb10* (Supplementary Table 5), suggesting that NR2F2 directly regulates its expression. Given the importance of *Tmsb10* in the differentiation of interstitial cells into fetal Leydig cells, the reduced *Tmsb10* expression observed in *Nr2f2* cKO testes indicates that not only are fewer interstitial cells present, but the remaining *Nr2f2*-negative cells may also have impaired capacity for Leydig cell differentiation. We have included this information in the introduction (Lines 83-84), results (Lines 160 and 247), and discussion sections of the manuscript (Lines 616-627).

5. A major unresolved issue is that while *Nr2f2* is hypothesized to suppress fetal Leydig cell differentiation, its knockout leads to impaired differentiation. Although the authors propose various hypotheses based on ChIP-seq and transcriptome data, the precise mechanism remains elusive, leaving the reader with an incomplete understanding.

Response: We understand this reviewer's sentiment. To clarify reviewer's concern, we have done the following. Based on the transcriptomic analysis followed by immunofluorescence confirmation, we discovered a general reduction in interstitial cell proliferation in knockout gonads compared to controls (Supplementary Fig. 6B-C). Furthermore, within the knockout gonads, cells lacking *Nr2f2* exhibited lower proliferation rates than those retaining *Nr2f2* (Supplementary Fig. 6D). These findings suggest that *Nr2f2* is essential for interstitial cell proliferation, which likely contributes to the observed reductions in testicular size (Supplementary Fig. 4B), interstitial area (Supplementary Fig. 4C), and interstitial cell numbers (Supplementary Fig. 4E). We have included these results in the text (lines 321-330). Moreover, thanks to the reviewer suggestions we noticed that *Nr2f2* cKO resulted in a downregulation of *Tmsb10*, a gene involved in Leydig cell differentiation. The incorporation of this information in our model could explain why there is a reduced Leydig cell differentiation when *Nr2f2* is knocked out (See comment 6). Lastly, our multiomics analysis revealed that Leydig cell differentiation does not rely on changes in chromatin accessibility but rather on the activity of interstitial and Leydig cell-specific transcription factors within pre-accessible chromatin regions. This suggests that, although NR2F2 is not directly occupying these regions to inhibit differentiation, a positive regulatory factor is required to activate them. Among the candidates, we identified several key transcription factors previously implicated in Leydig cell differentiation, including NR5A1, GATA6, and ESRRG. We believe this proposed mechanism integrates both existing and new data, offering a more comprehensive understanding of Leydig cell differentiation. As we have generated a *Nr2f2-CreER* model that allows gene targeting in fetal Leydig progenitor cells, we plan to test some of these hypotheses by knocking out specific Leydig cell transcription factors in progenitor cells in the future.

7. On a minor note, in the schematic figure (Fig. 7), fetal Leydig cells are depicted with

cellular polarity, which seems inappropriate. Fetal Leydig cells are interstitial and do not typically exhibit unidirectional cellular projections.

Response: Figure 7 is now updated with a non-polarized cell.

Reviewer #3 (Remarks to the Author):

Main comments:

1. Regarding the terminology for Leydig progenitor cells, the term "non-steroidogenic interstitial cells" is too broad, as various somatic cells could fall under this category. Additionally, although these progenitors are not themselves steroidogenic, they belong to a lineage where the differentiated Leydig cells primarily perform steroidogenesis. A more accurate and descriptive term would be "steroidogenic progenitors" or "Leydig cell progenitors", as this better reflects their role in the cell lineage.

Response: We agree that the terminology is not well defined in the field. For clarity when we talk about c0 we will rename them as "steroidogenic progenitors" to avoid confusion.

2. It would be beneficial to mention, either in the introduction or discussion, the results of a recent study on murine steroidogenic lineages, that combined single-cell RNA sequencing (scRNA-seq) and lineage tracing (Ademi et al., 2022). This study showed that steroidogenic progenitors differentiate into fetal Leydig cells (FLCs) within a narrow developmental window, specifically around embryonic days E12.5 and E13.5. By E16.5, these progenitors predominantly give rise to peritubular myoid cells, while the remaining population later differentiates into adult Leydig cells (ALCs). In addition, the study identified a transient population of pre-Leydig cells at E12.5 and E13.5, which were characterized transcriptionally.

Response: We now included a section in the discussion regarding the findings from Ademi 2022 (Lines 525-533).

3. The first section of the manuscript, which describes the characterization of interstitial cell populations in the embryonic testis, is crucial but would benefit from further analysis. Given the significant plasticity of steroidogenic progenitors and the presence of intermediate cell types at E12.5 and E13.5 (e.g. pre-LCs), it is recommended that the multiomic single-cell analysis be extended to earlier developmental stages, particularly E11.5, E12.5, and E13.5, to provide a more comprehensive view of the lineage differentiation processes. The E14.5 stage is too late to assess the early stages of steroidogenic progenitor differentiation into FLCs. This dual approach would provide insight into the regulatory mechanisms driving the transition from progenitor cells to fetal Leydig cells (FLCs) and the emergence of other intermediate cell types.

Response: We did perform the single-nucleus multiomic analysis on additional developmental time points such as E11.5, E12.5, and E13.5; however, the analyses of these additional time frame are included in other submission. We believe that even without these earlier time points, the E14.5 single cell analysis and the lineage tracing experiment together provide strong evidence to demonstrate that *Nr2f2*-positive cells give rise to fetal Leydig cells.

4. On lines 296-308, the authors characterized the interstitial compartment of the mouse testis using single-nucleus multiomics data, and identified five distinct clusters. They confirmed the annotation and localization of different cell populations through immunofluorescence, specifically detecting ALDH1A2+ testicular epithelial cells, ITGA8+ mesonephric cells, and LGI1+ pericytes. These findings are significant and should be included in one of the primary figures in the manuscript. It is important to note that MYLK1 is expressed in several groups of interstitial cells, so it should not be considered as a specific marker for group 1. Furthermore, the expression profiles of *Acta2* and *Myh1* suggest that cluster 1 consists predominantly of peritubular myoid cells.

Response: As suggested, we now move these figures from supplementary data to Fig. 1C and D and replaced the MYLK1 for ACTA2 as a marker for the population. We re-named c1 as “contractile cells” as we believe is comprised of both peritubular myoid cells and tunica cells.

5. As noted in lines 326-329, *Nr2f2* is highlighted in the differential expression analysis and used to lineage trace steroidogenic progenitors; however, there is no UMAP or any figure illustrating the expression of *Nr2f2* in the single nuclei data. Including such a visual representation would enhance the clarity and impact of the findings regarding the role of *Nr2f2* in these progenitors.

Response: We now included a feature plot showing *Nr2f2* expression (Supplementary Fig. 2A).

6. The authors used *Nr2f2*-CreER;CAGSun1/sfGFP reporter mice to track the fate of NR2F2+ cells in the mouse testis in an inducible manner. Samples were collected at E14.5 following three different tamoxifen (TAM) injection protocols: 1) one injection at E11.5, 2) one injection at E12.5, and 3) one injection at E13.5. While the strategy to follow the fate of NR2F2+ cells at E14.5 is relevant, we strongly recommend that the authors evaluate recombination efficiency 24 hours after each injection, as they did not report the proportion of NR2F2+ cells targeted by their transgenic system. The low proportion of GFP+ cells within the NR2F2+ population 24 hours after TAM injection at E13.5 highlights a major limitation of the system, which appears unsuitable for labeling a substantial proportion of NR2F2+ cells after a single injection.

Response: It is well established that three consecutive tamoxifen injections are typically recommended for optimal cell labeling with the CreER system. However, the aim of our

experiments was not to label all cells, but rather to label specific cell populations at particular time points and track their fate, a well acceptable scheme for cell fate mapping in the field. Consequently, we did not intend to label all cells at any of the evaluated time points. Given this, we believe the number of labeled cells we achieved with a single tamoxifen injection exceeded our expectations, especially in comparison to other CreER systems. Additionally, we observed that the co-staining with NR2F2 in our images may have obscured the true proportion of labeled cells. To address this, we now include a supplementary figure (Supplementary Figure 3A), where we show only the GFP-Sun1 channel from our four lineage tracing images. We also noticed that the original E13.5 image was presented at a different magnification and orientation than the rest. We have now corrected this and provided the image at the appropriate magnification in Fig. 3D. Among the samples, the E12.5 sample showed the lowest labeling, but since labeling at E13.5 and E14.5 appeared more robust, we attribute this variability to differences in tamoxifen injection efficiency.

7. The authors claim that cluster c0 represents progenitors of cluster c3 at E14.5; however, their lineage tracing was performed with inductions at E11.5, E12.5, and E13.5, all preceding E14.5. This temporal discrepancy complicates the conclusion that the observed differences in transcriptomes and chromatin accessibility between c0 and c3 accurately reflect the processes involved in fetal Leydig cell differentiation. To improve the characterization of the progenitors associated with the lineage tracing experiments, it would be beneficial to generate multiomic data at the appropriate developmental time points.

Response: In our original submission, we included lineage tracing data performed at E14.5 and evaluated at E17.5, which was initially presented as a supplementary figure. However, we have now moved this result to the main figure (now Fig. 3E). When *Nr2f2*-positive cells were labeled at E14.5, we observed SUN1-GFP-positive fetal Leydig cells at E17.5. This suggests that *Nr2f2*-positive steroidogenic progenitor cells (c0) at E14.5 are capable of differentiating into Leydig cells (c3), consistent with the findings from our E14.5 multiomics data. We believe that additional multiomics analysis would not provide further insights beyond what is already revealed by our lineage tracing data.

8. It is not possible to conclude that differentiation and the production of fetal Leydig cells (FLCs) is a continuous process during testis development, as suggested in lines 361, 366, and 643. Induction at E11.5, E12.5, and E13.5 alone is not sufficient to support this conclusion. Lineage tracing studies (Ademi et al 2022) have shown that the vast majority of both FLCs and ALCs are derived from steroidogenic progenitors that are primarily specified around E11.5. Only a small fraction of steroidogenic progenitors is specified later, and they contribute minimally to the overall FLC and ALC populations.

Response: In our original submission, we included lineage tracing data performed at E14.5 and evaluated at E17.5, which was initially presented as a supplementary figure. To address reviewer's comment, we have now moved this result to the main figure (now Fig.

3E). When *Nr2f2*-positive cells were labeled at E14.5, we observed SUN1-GFP-positive fetal Leydig cells at E17.5. This suggests that *Nr2f2*-positive steroidogenic progenitor cells (c0) at E14.5 are capable of differentiating into Leydig cells (c3), which is consistent with our E14.5 multiomics data. If fetal Leydig cells were specified at E11.5, as suggested by *Ademi et al.*, we would expect to observe no labeled Leydig cells in our E12.5, E13.5 or E14.5 *Nr2f2* lineage tracing experiments, unless these specified steroidogenic progenitors (pre-Leydig cells) are *Nr2f2*-positive. Using our E14.5 single-nucleus multiomics data, we examined markers associated with the specification of pre-Leydig cells, as described in *Ademi et al.*, to determine whether these pre-Leydig cells are present at E14.5 (see below). Genes like *Aldh1a1* and *Cyp26b1* are expressed only in fetal Leydig cells, while others like *Rlim* and *Robo2* are expressed in various interstitial populations and/or other cell types. Among the genes analyzed, *Hhip* and *Osr2* appear to be specific to the c0 cluster, while *Bst2*, *Hsd11b2*, and *Sct* are expressed at low levels across several interstitial cell populations. These results did not provide consistent evidence for a distinct progenitor population at E14.5. What we believe is that *Ademi et al.* were detecting the first steroidogenic progenitor cells differentiating into Leydig cells.

From our findings, we conclude that *Nr2f2*-positive cells, potentially from the c0 cluster, serve as a pool of progenitors for fetal Leydig cell differentiation. Based on our lineage tracing results at E11.5, E12.5, E13.5, and E14.5, we propose that fetal Leydig cell differentiation from a *Nr2f2*-positive, non-steroidogenic population is a continuous

developmental process that occurs from at least E11.5 to E14.5. We have included these conclusions in the discussion section of our manuscript (Lines 525-533).

9. We suggest that the authors invert Figures 2 and 3 to improve the readability of the manuscript. Presenting the NR2F2 localization in the developing mouse gonad first would provide a clearer context before moving on to the lineage tracing experiments that follow the fate of NR2F2+ putative steroidogenic progenitors.

Response: Figures 2 and 3 are now reverted in order as suggested. We modified the text accordingly.

10. Concerns regarding the use of the *Wt1:CreERT* transgenic line: The tamoxifen-inducible Cre system is commonly used to control Cre-mediated recombination at specific times and in specific cell types. However, tamoxifen is not a neutral agent in this process; it is a well-documented endocrine disruptor known to cause adverse effects on the testis and the reproductive endocrine system (see doi: 10.1038/s41598-017-09016-4). In addition, the *Wt1:CreERT* transgene is a knock-in, which could lead to *Wt1* haploinsufficiency. When combined with tamoxifen exposure, this could lead to alterations in testicular development. It is important to know whether the authors included adequate controls to account for these factors and what they were?

Response: We acknowledge the limitations of using the *Wt1-CreERT2* model and carefully considered the potential effects when designing the experiments and analyzing the data. To control for any potential tamoxifen effects, both our control and *Nr2f2* cKO mice were obtained from the same pregnant mothers (littermates) and were injected with tamoxifen under identical conditions. Given that our control mice lack the *Wt1-CreERT2* allele, we aimed to assess whether the Cre alone or Cre combined with tamoxifen could mimic the phenotype observed in the *Nr2f2* cKO due to potential haploinsufficiency. As a positive control, our *Wt1CreERT2 Sun1* lineage tracing experiment shows that double tamoxifen injections at E11.5 and E12.5 in *Wt1-CreERT2* mice do not disrupt testicular morphogenesis (Supplementary Fig. 3B) and do not replicate the smaller testicles seen in the *Nr2f2* cKO mice.

11. We also recommend that the authors use the lineage tracing strategy in the NR2F2 cKO model to detect ARX/LacZ and confirm the NR2F2-independent expression of ARX. In addition, the detection of LacZ/HSD3B1 could help to determine whether NR2F2 is essential for Leydig cell differentiation. This would strengthen the conclusions regarding the role of NR2F2 in the fate of steroidogenic progenitors.

Response: We have now included a figure showing the co-expression of ARX and LacZ or NR2F2 in E14.5 control and *Nr2f2* cKO samples (Supplementary Figure 4A). However, since LacZ expression is driven by the *Nr2f2* promoter, only cells that normally express *Nr2f2*, such as ARX-positive or PDGFRA-positive interstitial cells, will

show LacZ expression. As *Nr2f2* is not expressed in fetal Leydig cells, these cells do not exhibit LacZ expression.

12. Interstitial progenitors have previously been shown to be associated with both the steroidogenic lineage and the peritubular myoid cell (PMC) lineage. However, the authors conclude that "a severely affected population in the *Nr2f2* cKO testes was the smooth muscle cells, particularly those in the tunica albuginea" (Lines 676-677). Given that the study focuses primarily on testis development up to E14.5, a stage before the full differentiation of PMCs, it would be insightful for the authors to evaluate the contribution of NR2F2+ cells to PMC differentiation. This could be achieved by co-labeling GFP with SMA at E17.5, when PMCs are more fully differentiated, to assess the role of NR2F2+ cells in PMC development.

Response: We have now included a figure evaluating the effects of *Nr2f2* cKO at E17.5 in the tunica and peritubular myoid cells (PMCs) (SMA-positive). Notably, the tunica is absent at E17.5 (Supplementary Fig. 7B), which is consistent with our E14.5 results (Fig. 4C). This suggests that *Nr2f2* is essential for tunica cell differentiation. Surprisingly, SMA-positive PMCs, which are also positive for LacZ or NR2F2, are present surrounding the testicular cords. This indicates that *Nr2f2* is not required for PMCs differentiation. However, we observed that the testicular cords are not properly shaped, which may suggest that the *Nr2f2* knockout affects PMCs function, potentially impairing their role in shaping the testicular cords. To assess whether *Nr2f2*-positive cells give rise to the tunica and PMCs, we performed lineage tracing at E11.5 (before PMCs and tunica cells differentiate in the gonad) and co-localized them with SMA at E17.5 (Supplementary Fig. 7C). We found that labeling *Nr2f2*-positive cells at E11.5 leads to the detection of GFP-positive cells in both the SMA-positive tunica and SMA-positive PMCs. Unfortunately, the strong tunica SMA staining makes it difficult to clearly visualize the staining in PMCs. This is included in a new section in the manuscript (Lines 369-390)

13. The term "bulk deconvolution" is incorrectly used in the manuscript. What the authors describe is not true deconvolution but rather an analysis of differentially expressed genes (DEGs) between cKO and WT which are then visualized across different single nucleus (SN) clusters. This is better described as an "empirical projection" rather than deconvolution. True deconvolution involves constructing reference signature matrices from single-cell data and projecting the bulk data onto these matrices using various statistical frameworks to estimate cell type proportions (see <https://doi.org/10.1038/s41467-020-19015-1> for more details). A more rigorous deconvolution approach could provide more robust insights into potential shifts in cell population proportions. This is particularly relevant in the context of bulk RNA-seq, as it allows for a more accurate representation of changes in cellular composition.

Response: We changed the bulk deconvolution to "empirical projection", as suggested.

14. Regarding the bulk RNA seq analysis (line 465), I have concerns about how the term "most affected population" is defined. This assessment cannot be reliably made visually on a heatmap, where color variations are relative to gene expression changes across the nuclei. Such an analysis only indicates that certain genes expressed in specific subpopulations are differentially expressed in the bulk, which lacks rigor. A more informative approach would be to quantify the contribution of each gene to each cluster and then to calculate the effect of the observed expression variations in differentially expressed genes (DEGs). The authors should exercise caution in making such claims.

Response: We thank the reviewers for this insightful comment. In response, we have now included a quantitative definition for identifying the cell populations predicted to be "most affected" (Lines 306-310). We reason that cell populations with relatively higher expression of the differentially expressed genes (DEGs) are more likely to experience significant changes due to the loss of *Nr2f2*. Therefore, we define the "most affected" populations as those where more than 20% of the DEGs have a z-score greater than 1 across the testicular cell populations in the snRNA-seq data. This data is now included in Supplementary Table 4.

15. While gene groups can to some extent be associated with specific populations, their up or down-regulation in the cKO can also be linked to a reduction in population size rather than an alteration in its transcriptome. Standard bulk deconvolution utilizing analytical tools like MUSiC could yield valuable insights but would eventually fail at clarifying what is really changing in the cKO: population proportions or their transcriptome. Manual quantification of the possibly affected populations or performing single-nucleus multiome analyses on cKO testes would provide a clearer understanding of the underlying changes in the mutant model.

Response: We believe that single-cell analysis of the *Nr2f2* cKO testes would provide information similar to our bulk RNA-seq and sequential "empirical projection." Our bulk RNA-seq and sequential "empirical projection" identified and validated the cell types that were affected in *Nr2f2* cKO samples and the potential pathways affected on each cell type. Additionally, we quantified, as suggested, both interstitial cells and Leydig cells at E14.5 (Supplementary Figure 4) and also Leydig cells at E17.5 (Fig. 5), which show a reduction in cell numbers in the gonad. One of the pathways identified in our empirical projection was the G2/M transition. To investigate this further, we quantified the proliferative status of interstitial cells in both control and knockout gonads. Our data show a significant reduction in interstitial cell proliferation in the knockout compared to the control gonads (Supplementary Fig. 6B & C). Moreover, in the knockout gonads, cells lacking *Nr2f2* proliferated less than those where *Nr2f2* remained intact (Supplementary Fig. 6D). These findings suggest that *Nr2f2* is essential for interstitial cell proliferation. This could explain the observed reductions in testicular size (Supplementary Fig. 4B), testicular interstitium area (Supplementary Fig. 4C), interstitial cell number (Supplementary Fig. 4E), and possibly impaired Leydig cell differentiation (Supplementary Fig. 4G) due to a decreased pool of progenitor cells.

16. Single-cell and ChIP-seq data should be accompanied by the QC plots.

Response: We now include the QC plots for our data (Supplementary Figure 10 and 8, respectively)

17. ChIP-seq should be accompanied by metaplots and heatmaps of the peak coverage profile/distribution to see the quality and pattern of the signal to objectively evaluate the quality of the data (for ex., DOI: 10.26508/lsa.202201854, Fig 1).

Response: We now include the suggested data in Supplementary Figure 8

Minor Comments:

1. Line 79: for Wnt5a+ cells, this also includes adult Leydig cells

Response: We are aware that Wnt5a and Tcf21 cells give rise to adult Leydig cells, but as we are only focusing on embryonic stages, we decided to omit that detail for clarity.

2. Line 156: Transgene names should be italicized.

Response: This is now corrected in the manuscript.

3. Line 166: Start the sentence with "Five" instead of using the number "5".

Response: Corrected.

4. Line 186: Clarify what "0.3% acid alcohol" refers to (e.g., specify the composition of the solution).

Response: We clarified that section.

5. Lines 296-308: The authors characterized the interstitial compartment of the mouse testis using single-nucleus multiomics data, and identified five distinct clusters. They confirmed the annotation and localization of different cell populations through immunofluorescence, notably detecting ALDH1A2+ testicular epithelial cells, ITGA8+ mesonephric cells, and LGI1+ pericytes. These findings are significant and should be integrated into one of the primary figures in the manuscript. MYLK1 is expressed across several groups of interstitial cells, so it should not be regarded as a specific marker for Group 1. Furthermore, the expression profiles of Acta2 and Myh1 suggest that cluster 1 predominantly consists of peritubular myoid cells.

Response: As suggested, we moved these figures from supplementary to Fig. 1D&E and replaced the MYLK1 for ACTA2 as a marker for the population. We re-named c1 as

“contractile cells” as we believe is comprised of both peritubular myoid cells and tunica cells.

6. Line 405: Replace Fig S3B by Fig S4B

Response: Corrected

7. Line 419: Replace muscle cells with contractile cells

Response: Corrected

8. Line 515: Replace "majority" with "the biggest part".

Response: Corrected

9. Regarding Tables S6, to improve readability and usability, I recommend adding descriptive titles directly within the document. Additionally, please ensure that the filenames of the tables are consistent and clearly labeled. Furthermore, some column names in Table S6 lack clarity, making it challenging to infer the data presented.

Response: Tables are updated to improve the clarity

10. Lines 656-657: The statement is inaccurate because the characterization of the different cell populations in the testis using a gonadal atlas with scRNA sequencing was already published two years ago. This study described two populations of endothelial cells and identified a population of immune cells.

Response: We remove that claim as suggested.

Reviewer #2

1. In Supplemental Figure 2C, the results of the trajectory analysis are presented. According to this figure, C0 is suggested to be the progenitor of C3 (fetal Leydig cells), but a differentiation pathway from C1 to C0 is also observed. Since C1 expresses *Mylk*, *Myh11*, and *Acta2*, it can be identified as a contractile cell. Considering this, along with the disappearance of the tunica albuginea observed in *Nr2f2*-cKO mice, could it be interpreted that the reduction in the supply of fetal Leydig progenitors from the tunica albuginea leads to a decrease in both interstitial cells and fetal Leydig cells? In fact, many of the genes whose expression is reduced in *Nr2f2*-cKO mice appear to be associated with ECM and contractile function. If the above-mentioned mechanism can also be considered, even partially, alongside the gene expression-mediated mechanism proposed by the authors, how about incorporating it into the discussion?

Response: We agree with the reviewer that the c0 and c1 are connected by a trajectory in our trajectory analysis. Based on the pseudo-time analysis, c0 seems to contribute to c1 and not the other way around. Moreover, the lineage tracing experiment also suggests that c0 contributes to both Leydig and tunica cells. We included the trajectory analysis and pseudo-time information in the results section (Lines 402-404). We also included this information in the discussion (Lines 600-607). These results together implicate that c0 acts as a progenitor for both c1 and c3. As *Nr2f2* is expressed in tunica cells, we cannot elucidate if the lack of tunica cells in the *Nr2f2* cKO is due to lower number of progenitor cells, like in the case of the fetal Leydig cells, or if *Nr2f2* is required to trigger tunica cell differentiation. We don't have solid evidence to suggest that c1 contributes to c0 or c3.

2. For testosterone measurements in fetal testes, especially in the small testes of cKO mice, I believe that mass spectrometry is the most appropriate method. While I am unable to provide specific names, there are researchers and companies that measure steroid hormones using mass spectrometry on blood or tissue samples. To explain the dramatic phenotype observed in the external genitalia, showing only the number of Leydig cells seems insufficient. To demonstrate the functional capacity of the remaining Leydig cells compared to normal Leydig cells, testosterone measurement would be ideal. If this is challenging, histological analyses could serve as supplementary data. Observing lipid droplets and rough endoplasmic reticulum using electron microscopy would be ideal, but if this is also difficult, perhaps optical microscopy-level observations or Oil Red O staining could be attempted.

Response: Unfortunately, we were not able to detect testosterone using mass spectrometry in our embryonic samples, even in the control testis, potentially due to the small testicular size. We therefore followed the suggestion of the reviewer and performed ORO staining on control and *Nr2f2* cKO testis (Supplementary Fig. 7B). *Nr2f2* cKO gonads had a reduced number (consistent with the lower number of fetal Leydig cells) and less intense ORO positive lipid droplets (Supplementary Fig. 7B), suggesting a defective steroidogenesis. We included this new information in the results section (Lines 377-381) and in the discussion (Lines 663-665).

Reviewer #3

Before addressing the two remaining issues, I would like to underscore some quantitative and temporal information regarding steroidogenic progenitors and Leydig cells that are currently missing from the manuscript's introduction but are crucial for providing context. These information should be incorporated into the introduction in some form. In mice, the first Leydig cells appear toward the end of E12.5, with their numbers increasing substantially from E13.5 onwards.

Response: We agree with this timeline, and we include it now in the introduction (Lines 61-62)

This timeline suggests that the *Nr2f2*⁺/*Wnt5a*⁺/*Arx*⁺ steroidogenic progenitors at the origin of fetal Leydig cells must be specified during an earlier developmental window. This observation is supported by studies combining scRNA-seq and quantitative lineage tracing, which demonstrated that *Wnt5a*⁺/*Nr2f2*⁺/*Arx*⁺ steroidogenic progenitors specified between E11.5 and E12.5 contribute to the majority of fetal Leydig cells (74%) and adult Leydig cells (79%) (Ademi et al 2022). These findings indicate that the majority of both fetal and adult Leydig cells originate from *Wnt5a*⁺/*Nr2f2*⁺/*Arx*⁺ steroidogenic progenitors primarily specified around E11.5–E12.5. In contrast, only a minor fraction of steroidogenic progenitors are specified later, and these contribute minimally to the overall populations of fetal and adult Leydig cells.

Response: The lineage tracing experiment by Ademi et al in 2022 was done by labelling *Wnt5a*⁺ cells at E11.5 and E12.5, leading to the conclusion that *Wnt5a* positive cells between E11.5 and E12.5 gave rise to majority of fetal and adult Leydig cells. But whether *Wnt5a* progenitors are able to continuously give rise to FLC after E12.5 remains unknown. Our lineage tracing experiments at E12.5, E13.5 and E14.5 provide important information that *Nr2f2* positive cells retain progenitor cell activity during these time periods. We therefore propose that progenitor-to-fetal Leydig cell differentiation is a continuous process, occurring, at least, between E11.5 and E14.5, from a pool of *Nr2f2*-positive progenitor cells in the interstitium. We expanded in the discussion (Lines 547-562)

Comment 3: The first two chapters of the results section are critical in establishing context and characterizing the interstitial populations of the testis, particularly the *Nr2f2*⁺ steroidogenic progenitors. A significant concern that remains inadequately addressed pertains to the specification of these steroidogenic progenitors and their considerable plasticity during the critical developmental window between E11.5 and E13.5. By focusing solely on the E14.5 stage, the manuscript misses key insights into how *NR2F2*⁺ steroidogenic progenitors are initially specified, their transcriptional evolution, and their subsequent differentiation into fetal Leydig cells. Expanding the analysis to include earlier stages, particularly E12.5 and E13.5, would greatly enhance the understanding of these key developmental processes. Given that multi-omic data for these earlier stages are available, I strongly recommend incorporating scRNA seq data from E12.5 and E13.5 to comprehensively analyze the emergence and evolution of steroidogenic progenitors between E12.5 and E14.5. Including these data would substantially enhance the depth and overall impact of the study, especially for a manuscript intended for publication in Nature Communications.

Response: As suggested by the reviewer, we combined the E12.5 and E13.5 and E14.5 multiome data (Supplementary figure 2F), and performed clustering and expression analysis. We found that *c2* is

the potential steroidogenic progenitors (*Nr2f2* positive) and c10 is the fetal Leydig cells (*Nr5a1* positive). Moreover, *Nr2f2* positive steroidogenic progenitor cells form a continuum with c10 fetal Leydig cells at all the evaluated timepoints (E12.5, E13.5 and E14.5). This suggests that *Nr2f2* positive steroidogenic progenitor cells differentiate to fetal Leydig cells at E12.5, E13.5 and E14.5. This is consistent with our lineage tracing experiments. Moreover, this also supports that progenitor-to-fetal Leydig cell differentiation is a continuous process, occurring, at least, between E12.5 and E14.5, from a pool of *Nr2f2*-positive progenitor cells in the interstitium. As the differentiation process from E12.5, E13.5 and E14.5 overlaps, we consider that the analysis of the progenitor-to-fetal Leydig cell differentiation that we performed using the single cell multiome data at E14.5 should reflect the differentiation process that also occurs in early developmental timepoints (E12.5 and E13.5). We included this new analysis in lines 163-172.

Comments 2, 8: There seems to be a tendency to underrepresent or overlook prior contributions from other laboratories. Ensuring accurate and comprehensive citation of existing work is crucial, as it does not diminish the value of the authors' contributions in this study but rather contextualizes their findings within the broader field. While the authors incorporated findings from Ademi et al. (2022) into the revised manuscript, as suggested in the initial review (Lines 525–533), the current phrasing is inaccurate and could be improved. The conclusions of Ademi et al. are supported by multiple lineage-tracing experiments induced at various developmental stages. Additionally, there seems to be a misunderstanding in the rebuttal letter. The authors state: “If fetal Leydig cells were specified at E11.5, as suggested by Ademi et al.,...”. However, Ademi et al. did not claim that FLCs themselves are specified at E11.5. Rather, the study clarified that *Wnt5a*+/*Nr2f2*+/*Arx*+ steroidogenic progenitors—precursors to fetal Leydig cells—are specified during this period, and these progenitors subsequently differentiate into fetal Leydig cells from E12.5 onward. It would be beneficial for the readers to explicitly state in the introduction that the majority of both fetal and adult Leydig cells are derived from steroidogenic progenitors primarily specified around E11.5–E12.5.

We regret that the reviewer feels that we under-represent or overlook findings of others. This has never been our intention. We may have a different opinion and interpretation of others' finding, and we appreciate the reviewer for pointing out our differences. We completely agree with this reviewer that *Wnt5a* positive cells between E11.5 and E12.5 gave rise to majority of fetal and adult Leydig cells. However, similar to the paper by Ademi et al, we cannot claim that the steroidogenic progenitors are “specified” around E11.5-E12.5, as no lineage tracing was performed before (E10.5-E11.5). We expanded in the discussion (Lines 547-562). We agree that how *Nr2f2*+ progenitor cells are specified before E11.5 is important and interesting. Our future research will focus on answering this question. This current manuscript focuses on a different question and the results and conclusion will not be impacted without the information on E11.5 analysis.

Finally, I recommend adding a discussion of a recently published study (<https://www.biorxiv.org/content/10.1101/2024.07.17.602099v1>), which presents similar findings. Acknowledging this work would further contextualize the study's contributions and strengthen its impact.

We thank the reviewers for this suggestion, we have included this article in the manuscript as it become available in *elife* as a preprint (Lines 526-528 and 554-556).